EMBO
Molecular Medicine

# Efficacy of systemic temozolomide-activated phage-targeted gene therapy in human glioblastoma

Justyna Magdalena Przystal[1,†,‡], Sajee Waramit[1], Md Zahidul Islam Pranjol[1,§], Wenqing Yan[1], Grace Chu[1], Aitthiphon Chongchai[2], Gargi Samarth[1], Nagore Gene Olaciregui[3,4], Ghazaleh Tabatabai[5,6], Angel Montero Carcaboso[3,4], Eric Ofori Aboagye[7], Keittisak Suwan[1] & Amin Hajitou[1,*]

## Abstract

Glioblastoma multiforme (GBM) is the most lethal primary intracranial malignant neoplasm in adults and most resistant to treatment. Integration of gene therapy and chemotherapy, chemovirotherapy, has the potential to improve treatment. We have introduced an intravenous bacteriophage (phage) vector for dual targeting of therapeutic genes to glioblastoma. It is a hybrid AAV/phage, AAVP, designed to deliver a recombinant adeno-associated virus genome (rAAV) by the capsid of M13 phage. In this vector, dual tumor targeting is first achieved by phage capsid display of the RGD4C ligand that binds the $\alpha_v\beta_3$ integrin receptor. Second, genes are expressed from a tumor-activated and temozolomide (TMZ)-induced promoter of the glucose-regulated protein, *Grp78*. Here, we investigated systemic combination therapy using TMZ and targeted suicide gene therapy by the RGD4C/AAVP-*Grp78*. Firstly, *in vitro* we showed that TMZ increases endogenous *Grp78* gene expression and boosts transgene expression from the RGD4C/AAVP-*Grp78* in human GBM cells. Next, RGD4C/AAVP-*Grp78* targets intracranial tumors in mice following intravenous administration. Finally, combination of TMZ and RGD4C/AAVP-*Grp78* targeted gene therapy exerts a synergistic effect to suppress growth of orthotopic glioblastoma.

**Keywords** bacteriophage; glioblastoma; Grp78; targeting; temozolomide
**Subject Categories** Cancer; Genetics, Gene Therapy & Genetic Disease; Pharmacology & Drug Discovery

## Introduction

Glioblastoma, also known as glioblastoma multiforme (GBM) or grade IV astrocytoma, is the most common and deadliest of primary brain tumors in adults with 14.6-month median survival time and a 2% 5-year survival rate (Kwiatkowska *et al*, 2013). This depressing prognosis occurs with current, aggressive gold standard treatment regimens, such as chemotherapy. Even more recent trials with anti-angiogenic and molecularly targeted therapies have shown only slight improvements in GBM outcome (Gilbert *et al*, 2014). Therefore, novel remedies that effectively treat or improve/complement existing treatments are pressingly needed. Temozolomide (TMZ), an alkylating agent able to cross the blood–brain barrier (BBB), is well-tolerated and widely used as part of standard chemotherapy for newly diagnosed and recurrent GBM, but has limited efficacy (Thon *et al*, 2013; Messaoudi *et al*, 2015; Taal *et al*, 2015). We hypothesized that combining TMZ chemotherapy with gene therapy should yield a synergistic effect against GBM. Gene therapy for GBM has been attempted for the past 24 years, but progress has been hindered mostly by lack of tumor selectivity and inefficiency of eukaryotic viral vectors (Natsume & Yoshida, 2008; Kwiatkowska *et al*, 2013). Gene therapy of brain tumors is further hampered by the BBB (Maguire *et al*, 2014). Although this could be circumvented by delivery of vectors locally, the invasiveness of repeated intracerebral injections and the diffuse/not discrete nature of GBM place great limitations. We have introduced a unique prokaryotic viral-based approach of systemic intravenous gene delivery to target tumors specifically by using the harmless and non-pathogenic filamentous M13 bacteriophage, a virus that infects only bacterial cells and lacks native tropism to normal tissues in human and eukaryotes in general (Asavarut & Hajitou, 2014). A key strategy of our

1 Phage Therapy Group, Division of Brain Sciences, Department of Medicine, Imperial College London, London, UK
2 Thailand Excellence Centre for Tissue Engineering and Stem Cells, Department of Biochemistry, Faculty of Medicine, Chiang Mai University, Chiang Mai, Thailand
3 Institute de Recerca Sant Joan de Deu, Barcelona, Spain
4 Department of Pediatric Hematology and Oncology, Hospital Sant Joan de Deu, Barcelona, Spain
5 Interdisciplinary Division of Neuro-Oncology, Hertie Institute for Clinical Brain Research, Center for CNS Tumors, Comprehensive Cancer Center, University Hospital Tübingen, Eberhard Karls University, Tübingen, Germany
6 German Cancer Consortium (DKTK), DKFZ Partner Site Tübingen, Tübingen, Germany
7 Comprehensive Cancer Imaging Centre, Imperial College London, Faculty of Medicine, London, UK
*Corresponding author. Tel: +44 207 594 6546; E-mail: a.hajitou@imperial.ac.uk
†Present address: Interdisciplinary Division of Neuro-Oncology, Hertie Institute for Clinical Brain Research, University Hospital Tübingen, Eberhard Karls University, Tübingen, Germany
‡Present address: German Cancer Consortium (DKTK), DKFZ Partner Site Tübingen, Tübingen, Germany
§Present address: William Harvey Research Institute, Queen Mary University of London, London, UK

systemic delivery platform involves the construction of a hybrid bacteriophage vector termed AAV-phage, or AAVP (Fig EV1A). In this vector, the single-stranded genome of human adeno-associated virus (AAV) was inserted within the M13 phage single-stranded genome, resulting in a phage capsid which incorporates AAV genomes(s) (Hajitou *et al*, 2006, 2007). Importantly, the phage capsid was engineered to display the CDCRGDCFC (RGD4C) ligand (Fig EV1A) that binds the heterodimer $\alpha_v\beta_3$ integrin cell surface receptor, which is overexpressed on tumor cells and supporting angiogenic vasculature in most tumor types including human GBM (Hajitou *et al*, 2006; Schnell *et al*, 2008). Upon binding to $\alpha_v\beta_3$ integrin receptor and subsequent entry of the RGD4C/AAVP viral particles into cells, the AAV genome is released to express genes in tumors from a cytomegalovirus, *CMV*, promoter (Hajitou *et al*, 2006). To date, we and collaborators have established that these vectors target various preclinical models of human cancer including soft tissue sarcoma, melanoma, breast, prostate, and pancreatic cancers (Hajitou *et al*, 2006, 2008; Tandle *et al*, 2009; Yuan *et al*, 2013; Dobroff *et al*, 2016; Smith *et al*, 2016). Additionally, a study by the National Cancer Institute, USA, using our vector in pet dogs with natural cancers demonstrated that RGD4C/AAVP delivered the cytokine, tumor necrosis factor-alpha (*TNFα*), selectively, to spontaneous cancers (Paoloni *et al*, 2009). Remarkably, repeated vector dosing proved safe and resulted in complete tumor eradication in several dogs with aggressive cancers (Paoloni *et al*, 2009). To improve the vector platform for use in targeted gene therapy against GBM, we have refined the technology by replacing the *CMV* promoter with the tumor-specific *Grp78* promoter and designed the dual tumor targeting RGD4C/AAVP-*Grp78* vector (Kia *et al*, 2012, 2013). This novel vector effectively ensured further tumor selectivity, through transcriptional targeting as previously reported (Kia *et al*, 2012). We also demonstrated that the RGD4C/AAVP-*Grp78* vector provides much longer lasting transgene expression than the RGD4C/AAVP-*CMV* vector carrying a *CMV* promoter, *in vitro* and *in vivo* in subcutaneous GBM following intravenous administration (Kia *et al*, 2012). The *Grp78* promoter is marginally active in healthy tissues; however, potent activation has been observed in aggressive tumors, including GBM (Dong *et al*, 2004; Pyrko *et al*, 2007; Virrey *et al*, 2008), and is induced by stress and conditions of tumor microenvironment such as glucose deprivation, chronic anoxia, and acidic pH. It is further worth considering that the induction of the *Grp78* gene expression and activation confers drug resistance in a variety of human tumors, including gliomas (Li & Lee, 2006; Lee, 2007; Pyrko *et al*, 2007). *Grp78* can also be induced by TMZ in GBM (Pyrko *et al*, 2007), and its activation has been associated with GBM resistance to TMZ (Pyrko *et al*, 2007; Virrey *et al*, 2008). Thus, further increase in gene expression from the RGD4C/AAVP-*Grp78* can be ensured through TMZ activation of the *Grp78* promoter. Consequently, we postulated that RGD4C/AAVP-*Grp78* is a suitable candidate for use in combination with TMZ against GBM. Herein, we investigated the effects of combining TMZ chemotherapy and targeted gene therapy with RGD4C/AAVP-*Grp78*, termed chemovirotherapy, against intracranial orthotopic models of human glioblastoma. Firstly, we demonstrated that a list of various human glioblastoma cell lines as well as primary GBM and primary GBM stem cells express $\alpha_v\beta_3$ integrin, a receptor of RGD4C, resulting in ligand-directed gene delivery by RGD4C/AAVP *in vitro*. Then, we showed that TMZ treatment of human glioblastoma cell lines

increased expression of the endogenous *Grp78*, through the UPR (unfolded protein response) pathway, subsequently stimulating RGD4C/AAVP-*Grp78*-mediated gene expression. We also found that TMZ amplified the destruction of GBM cells in combination with RGD4C/AAVP-*Grp78-HSVtk* encoding the *Herpes simplex virus type I thymidine kinase* in the presence of ganciclovir (GCV); we used the *HSVtk* mutant SR39 (Black *et al*, 2001). Next, we confirmed that RGD4C/AAVP-*Grp78* targets orthotopic glioblastoma in mice after intravenous administration selectively binding to tumor cells and tumor vasculature without accumulation in the healthy brains. Additionally, the combination of TMZ and RGD4C/AAVP-*Grp78-HSVtk*/GCV administered systemically elicited strong anti-tumor activity against intracranial glioblastoma established *in vivo* from GBM cell lines and primary GBM, and in both immunodeficient and immunocompetent mice. Unless technically, the *in vivo* effect was measured synergistic, compared to TMZ or RGD4C/AAVP-*Grp78-HSVtk*/GCV alone. The HSVtk enzyme phosphorylates prodrug nucleoside analogues such as GCV and converts them into nucleoside analogue triphosphates. These triphosphate GCV compounds are then incorporated into the cellular genome, inhibit DNA polymerase, and subsequently induce cell death by apoptosis (Hamel *et al*, 1996; Natsume & Yoshida, 2008). It is important to note that the HSVtk/GCV approach also elicits a bystander effect, which means that cells containing the HSVtk kill neighboring non-transduced cells through the gap junctional intercellular communications (GJIC) that allow the transfer of the converted cytotoxic drug and/or toxic metabolites between these cells as we have previously reported (Trepel *et al*, 2009; Duarte *et al*, 2012). After a few days, "this bystander effect" results in increased cell killing by the RGD4C/AAVP-*HSVtk* vector and may potentially overcome the requirement for all malignant cells to be transduced in order to achieve meaningful tumor regression. Altogether, these findings indicate that this combination therapy strategy offers significant translational potential in the treatment regime for GBM patients.

## Results

### Human glioblastoma cell lines express integrin receptors of RGD4C and are subsequently transduced by the RGD4C/AAVP vector

As initial experiments, we conducted *in vitro* studies on cell lines by using three models of human glioblastoma cells, namely LN229, U87, and SNB19, considered as common cellular models of this disease. First, we investigated expression of the integrins $\alpha_v\beta_3$ and $\alpha_v\beta_5$, receptors for RGD4C/AAVP, by immunofluorescent staining of $\alpha_V$, $\beta_3$, and $\beta_5$ integrin subunits. As shown in Fig 1A, all tumor cells tested were positive for expression of $\alpha_v$, $\beta_3$, and $\beta_5$ integrins, with varying expression of each integrin. Next, we investigated RGD4C/AAVP-mediated gene delivery to these tumor cells and used vectors carrying the reporter *Luciferase* (*Luc*) gene, which allows quantitative analysis of gene expression, and analyzed *Luc* expression over time. Cells were incubated with targeted RGD4C/AAVP-*Luc* or control non-targeted/AAVP-*Luc* vector (lacking the RGD4C). RGD4C/AAVP-mediated gene expression was demonstrated in all the human glioblastoma cells tested, in an efficient way and which increased over time (Fig 1B). Importantly, gene expression

mediated by RGD4C/AAVP was selective, targeted, and dependent on RGD4C ligand binding to integrin receptors as no *Luc* expression was detected in cells treated with the control non-targeted/AAVP-*Luc* (Fig 1B).

### Temozolomide treatment of human glioblastoma cells results in induction of the endogenous *Grp78* gene and boosts gene expression from the RGD4C/AAVP-*Grp78*

Numerous studies have reported that treatment with chemotherapeutic drugs stimulates endogenous *Grp78* gene expression in tumors (Pyrko *et al*, 2007). It has also been reported that treatment with TMZ induces expression of endogenous *Grp78* in glioblastoma patients, which is associated with glioblastoma resistance to TMZ (Virrey *et al*, 2008). Thus, we examined whether TMZ can increase the *Grp78* promoter activity in the context of RGD4C/AAVP-*Grp78* vector *in vitro* and subsequently enhance gene expression. First, we sought to confirm that TMZ activates expression of the endogenous *Grp78* gene in glioblastoma cell lines. The LN229, U87, and SNB19 cells were treated with a range of TMZ concentrations (25–250 μM) for 12 h, followed by Western blot analysis and band intensity quantification. As shown in Fig 1C, the expression of endogenous *Grp78* increased in a dose-dependent manner in LN229, U87, and SNB19 cells. We also noted that *Grp78* expression in SNB19 reached a plateau between 100 and 250 μM of TMZ.

We next set out to determine the effect of TMZ on transgene expression mediated by the RGD4C/AAVP-*Grp78* in human glioblastoma cells. Vectors carrying the *Luc* reporter gene were utilized, and stably transduced LN229, U87, and SNB19 cells were generated by using *puro*® vectors that confer resistance to puromycin. Vectors expressing the *Luc* gene from the *CMV* promoter (RGD4C/AAVP-*CMV-Luc)* were also included in this experiment. Based on our data in Fig 1C, we used TMZ concentrations of 100 μM for LN229 and SNB19, and 60 μM for U87. Remarkably, TMZ induced a sharp increase in *Luc* expression in all LN229, U87, and SNB19 glioblastoma cells in which the *Luc* transgene was mediated by the RGD4C/AAVP-*Grp78-Luc* (Fig 1D). In contrast, *Luc* expression driven by the RGD4C/AAVP-*CMV-Luc* remained unaltered by TMZ in all tumor cells (Fig 1D). In LN229 and U87 cells, the rise in *Luc* expression occurred at 9 h post-TMZ treatment and continued gradually to reach 20- and 15-fold higher than RGD4C/AAVP-*CMV-Luc* at 12 h, respectively (Fig 1D). In the SNB19 cell line, the boost in *Luc* expression occurred at 24 h post-TMZ treatment and reached ~ 4.3-fold increase (Fig 1D).

To further investigate induction of the RGD4C/AAVP-*Grp78* vector in human glioma by other anti-cancer agents, known for their ability to cross the BBB, we treated human glioma cells with increasing concentrations of curcumin (0–40 μM). Curcumin, a natural and non-toxic dietary plant-based product, has been highlighted for its potential as an anti-cancer agent, due to its ability to induce tumor cell death with no systemic side effects. Studies have shown that curcumin can prevent proliferation of cancer cells including GBM (Klinger & Mittal, 2016). Addition of curcumin to tumor cells at day 3 post-transduction with RGD4C/AAVP-*Grp78-Luc* vector resulted in a dose-dependent increase of *Luc* gene expression that climbed markedly in the presence of 40 μM of curcumin (Fig EV1B).

### Temozolomide induces *Grp78* promoter activation through the UPR signaling pathway

To investigate whether the boost of RGD4C/AAVP-*Grp78* gene expression by TMZ is mediated through the conserved signaling cascade termed UPR (unfolded protein response) pathway that regulates the endogenous *Grp78* gene, we examined three transmembrane proteins representing the three arms of the UPR signaling pathway (Rutkowski & Kaufman, 2004; Pfaffenbach & Lee, 2011; Luo & Lee, 2013; Nagelkerke *et al*, 2014). These experiments were performed following TMZ treatment of LN229, U87, and SNB19 tumor cells stably transduced with the RGD4C/AAVP-*Grp78-Luc* vector, consistent with the experiments in Fig 1. We first focused on protein kinase RNA-like ER kinase (PERK); after homo-dimerization and trans-autophosphorylation, PERK phosphorylates Ser51 of the α-subunit of the eukaryotic translation initiation factor-2 (eIF2α; Luo & Lee, 2013; Ron & Walter, 2007) that selectively promotes translation of the transcription factor ATF4 which induces the *Grp78* promoter. Western blot analyses of tumor cells showed that TMZ induces expression of the phospho-eIF2α that is detectable at 2, 1, and 3 h post-TMZ treatment in LN229, U87, and SNB19 cells, respectively, and induction of expression persisted for up to 24 h post-TMZ treatment (Fig 2A). Next, we evaluated TMZ induction of expression of the activating transcription factor 6 (ATF6). We found that TMZ induces the ATF6-p90 expression detectable at 1 h in LN229 and 2 h in U87 cells, and then increased over time for 24 h post-treatment (Fig 2A). The induction of ATF6-p90 in SNB19 cells was noticed 24 h after TMZ treatment (Fig 2A). Previous studies have shown that an increase in the 90-kDa ATF6 level indicates initiation of the endoplasmic reticulum (ER) stress pathway and subsequently activation of UPR (Teske *et al*, 2011; Kia *et al*, 2012), which is a cellular stress response related to the ER stress. Finally, during ER stress, activated inositol-requiring enzyme 1 (IRE1) cleaves a 26-base intron from mRNA of X-box binding protein 1 (XBP1) resulting in a product that acts as a transcription factor for the *Grp78* promoter (Teske *et al*, 2011). Semi-quantitative RT–PCR analyses showed a slight increase in spliced XBP1 mRNA at 24 h post-TMZ treatment of LN229 (Fig 2B), while a greater and gradual increase in the level of spliced XBP1 mRNA was found in the U87 cell line upon TMZ treatment (Fig 2B). In contrast, no spliced XBP1 mRNA was detected in SNB19 glioblastoma cells (Fig 2B).

### Combination of TMZ with RGD4C/AAVP-*Grp78-HSVtk* and GCV produces strong tumor cell killing *in vitro*

Our previous studies showed that the RGD4C/AAVP-*Grp78-HSVtk* is superior to the conventional RGD4C/AAVP-*CMV-HSVtk* vector in inducing glioblastoma cell killing *in vitro* as well as GBM growth inhibition *in vivo* following intravenous administration of vectors carrying the *HSVtk*, and intraperitoneal injection of GCV (Kia *et al*, 2012). Herein, we compared tumor cell killing between two combinations of TMZ with RGD4C/AAVP-*Grp78-HSVtk*/GCV and TMZ with RGD4C/AAVP-*CMV-HSVtk*/GCV. We also included a comprehensive list of controls comprising cells treated with monotherapies of TMZ, RGD4C/AAVP-*Grp78-HSVtk* or RGD4C/AAVP-*CMV-HSVtk* vectors alone plus GCV, in addition to untreated cells. Cells stably transduced with RGD4C/AAVP-*Grp78-HSVtk* or RGD4C/AAVP-*CMV-HSVtk* were subjected to GCV or TMZ, or both for 4 days before

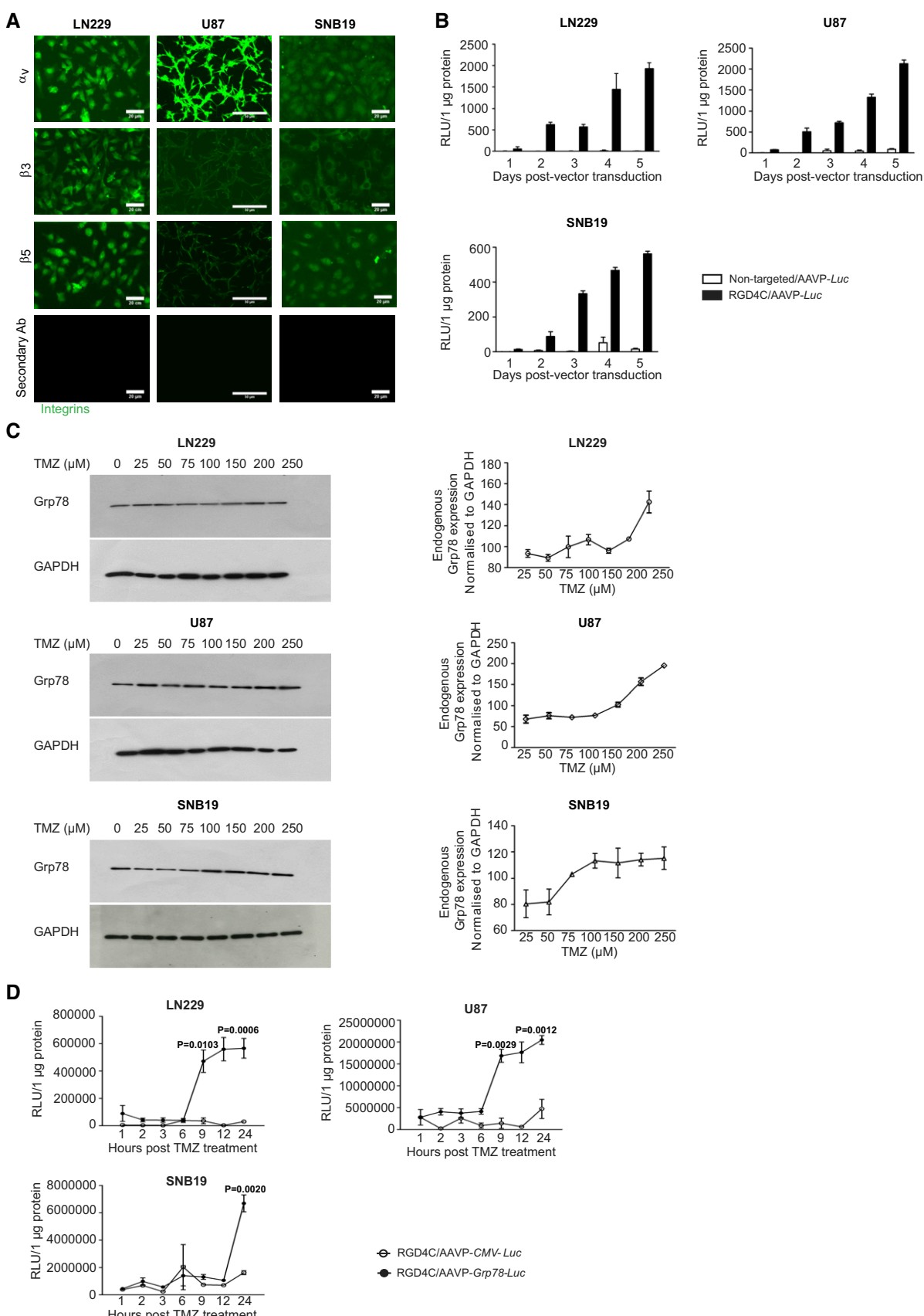

**Figure 1.**

**Figure 1. Targeted transduction of human glioblastoma cell lines and induction of RGD4C/AAVP-*Grp78* by TMZ.**

A Immunofluorescence staining of LN229, U87, and SNB19 tumor cells with primary antibodies against $\alpha_v$, $\beta_3$ or $\beta_5$ integrin subunits. Scale bars, 20 μm for LN229 and SNB19, and 50 μm for U87.

B Targeted gene delivery into LN229, U87, and SNB19 cells by RGD4C/AAVP-*Luc* carrying the *Luc* reporter gene. Cells were seeded into 48-well plates before transduction. Non-targeted/AAVP-*Luc* was used as negative control for targeted transduction. Results represent the average relative luminescence units (RLU)/1 μg protein. Data shown are representative of two experiments, $n = 3$.

C Western blot analysis of the endogenous Grp78 protein expression levels in LN229, U87, and SNB19 cells following treatment with increasing concentrations of TMZ. Grp78 level was normalized to GAPDH. Data shown in right panel are representative of two experiments, $n = 3$.

D Induction of *Grp78* promoter activity by TMZ in RGD4C/AAVP-*Grp78-Luc* cells. LN229, U87, and SNB19 cells stably transduced with RGD4C/AAVP-*Grp78-Luc* or RGD4C/AAVP-*CMV-Luc* were grown in the presence of TMZ for the indicated times. Results represent the average RLU/1 μg protein. Data shown are representative of two experiments, $n = 3$. Two-way ANOVA with Bonferroni correction (GraphPad Prism 6) was used for data analysis.

Data information: Data are expressed as mean ± SEM.
Source data are available online for this figure.

evaluation of the reactive oxygen species (ROS). Excessive production of ROS causes progressive oxidative damage and ultimately cell death. The current experiments have yielded numerous new findings. Firstly, both RGD4C/AAVP-*Grp78-HSVtk* and RGD4C/AAVP-*CMV-HSVtk* vectors showed cell killing in combination with GCV in all tumor cells (Fig 2C). However, the RGD4C/AAVP-*Grp78-HSVtk* generated greater glioblastoma cell killing, in general, than the RGD4C/AAVP-*CMV-HSVtk* (Fig 2C), which confirms our previous findings (Kia *et al*, 2012). Secondly, combination of RGD4C/AAVP/GCV with TMZ significantly decreased proliferation of all LN229, U87, and SNB19 glioblastoma cells, when compared to each RGD4C/AAVP/GCV or TMZ therapy alone (Fig 2C). Finally, importantly, cells treated with the combination of RGD4C/AAVP-*Grp78-HSVtk*/GCV and TMZ demonstrated significant increase in cell death compared to cells treated with combination of RGD4C/AAVP-*CMV-HSVtk*/GCV and TMZ (Fig 2C). Given these data and those on TMZ induction of gene delivery by RGD4C/AAVP-*Grp78-Luc* vector, we selected RGD4C/AAVP-*Grp78-HSVtk*/GCV as most suitable for *in vivo* studies and therapy combination with TMZ against orthotopic GBM.

**RGD4C/AAVP-*Grp78-HSVtk* vector targets intracranial human glioblastoma *in vivo* following intravenous administration**

These *in vivo* experiments were performed before initiating therapy studies in order to determine the ability of RGD4C/AAVP-*Grp78-HSVtk* vector to target an orthotopic model of human GBM *in vivo* following intravenous administration. As a preclinical model, we used brain tumors established in immunodeficient nude mice by intracranial implantation of U87 glioblastoma cells stably labeled with the firefly *Luc* transgene. Bioluminescent imaging (BLI) of *Luc* expression in tumors was used for tumor detection and for monitoring tumor growth. When tumors were established as revealed by BLI of *Luc*, we systemically (tail vein) administered RGD4C/AAVP-*Grp78-HSVtk* or control non-targeted/AAVP-*Grp78-HSVtk* ($5 \times 10^{10}$ transducing units (TU)/mouse). At 18 h after vector administration, tumor-bearing nude mice were killed, and then, tumors and control organs were collected. Immunofluorescence staining of frozen brain sections showed strong expression of $\alpha_v$ integrin in the tumors but not in the surrounding healthy brain (Fig 3A). Then, immunostaining of phage particles revealed distribution of the tumor-targeted RGD4C/AAVP-*Grp78-HSVtk* (green) throughout the tumor in the blood vessels (CD31—red) and surrounding cells (Fig 3B). No signal was detected in the surrounding healthy brain (Fig 3B).

Additionally, relative targeted RGD4C/AAVP-*Grp78-HSVtk* homing was quantified by recovery from tissue homogenates, bacterial infection, and counting the TU. We observed marked enrichment of RGD4C/AAVP-*Grp78-HSVtk* particles in orthotopic U87 xenografts (Fig 3C). Specifically, we found a ~ 12-fold enrichment compared with homing of non-targeted/AAVP-*Grp78-HSVtk* control to size-matched tumors and to control organs (Fig 3C). Furthermore, only background levels of RGD4C/AAVP-*Grp78-HSVtk* were observed in normal brains and control tissues, i.e., pancreas (Fig 3C). These data establish that RGD4C/AAVP-*Grp78-HSVtk* targets U87 orthotopic xenografts in nude mice upon intravenous administration. Moreover, further analyses of the vector tumor homing showed that we recovered an estimate of ~ $7 \times 10^6$ TU/g tumor tissue of intact RGD4C/AAVP particles following intravenous administration of a fixed vector dose of $5 \times 10^{10}$ TU/mouse. This does not take into account the non-infectious vector that has initiated processing by the cells (Fig 3D).

**Systemic chemovirotherapy combining RGD4C/AAVP-*Grp78-HSVtk*/GCV suicide gene therapy with TMZ enhances human glioblastoma destruction *in vivo***

Next, we evaluated therapeutic efficacy of the systemic chemovirotherapy *in vivo* against intracranial tumors, established by implantation of U87 cells labeled with the *Luc* reporter gene, and *Luc* expression in tumors was monitored by repetitive BLI of whole living tumor-bearing mice. We selected BLI of *Luc* expression as a simple way to monitor tumor viability and response to treatments (Hajitou *et al*, 2006, 2007, 2008; Kia *et al*, 2012; Przystal *et al*, 2013). At day 9 post-tumor cell implantation, mice with established intracranial U87 tumors (Fig 4A) received a single intravenous dose ($5 \times 10^{10}$ TU) of either targeted RGD4C/AAVP-*Grp78-HSVtk* or non-targeted/AAVP-*Grp78-HSVtk*. To optimize the model in the context of *in vivo* expression, TMZ was administered 4 days after intravenous vector delivery, as most of our previous studies (Hajitou *et al*, 2006, 2007; Kia *et al*, 2012; Przystal *et al*, 2013) have shown that initiation of RGD4C/AAVP-mediated gene expression in tumors occurs after day 4 following vector injection, at which time point the administered TMZ is able to boost *Grp78* promoter activity and subsequent gene expression. Furthermore, giving TMZ before initiation of gene expression could affect the intracellular trafficking of RGD4C/AAVP-*Grp78-HSVtk* and subsequent tumor cell transduction,

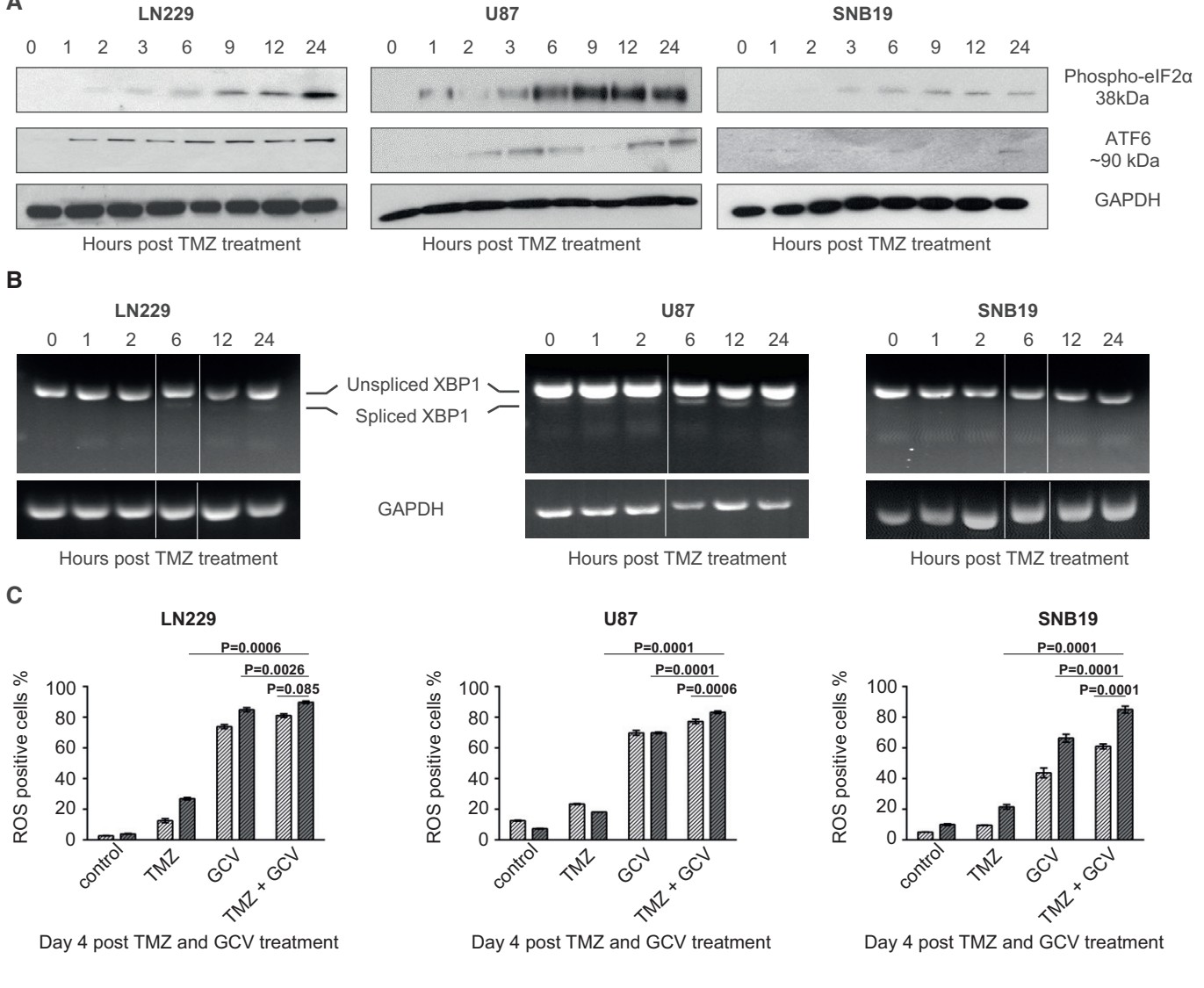

**Figure 2. TMZ activation of the UPR pathway and subsequent enhancement of glioblastoma cell killing *in vitro* by the RGD4C/AAVP-*Grp78-HSVtk* and GCV.**

A  TMZ induction of phospho-eIF2α and ATF6-p90 expression. Human glioblastoma cells transduced with RGD4C/AAVP-*Grp78* were analyzed by Western blot following treatment with TMZ (100 µM for LN229 and SNB19, 60 µM for U87). GAPDH was used as a control.

B  RT–PCR analysis of constitutive expression and splicing of XBP1 in glioblastoma cells transduced with RGD4C/AAVP-*Grp78*, in the presence of TMZ. Sizes of the PCR products were 289 bp for unspliced XBP1 and 263 bp for spliced XBP1, and the lower size band is not specific. Time points were selected among those tested in the Western blot in (A), subsequently the image containing 0, 1, and 2 h was juxtaposed to images of 6, 12, and 24 h. A white line has been added between the gel pieces that have been juxtaposed.

C  Tumor cell killing *in vitro* by the *HSVtk*/GCV approach. Glioblastoma cells stably transduced with RGD4C/AAVP-*Grp78-HSVtk* or RGD4C/AAVP-*CMV-HSVtk* were treated with either GCV (10 µM) or TMZ (100 µM for LN229 and SNB19, 60 µM for U87) or combination of both GCV and TMZ. Cells were stained with MitoSOX and analyzed by FACS at day 4 post-treatment. Data shown are representative of three experiments, *n* = 3. Two-way ANOVA with Tukey's multiple comparison test (GraphPad Prism 6) was used for data analysis.

Data information: Data are expressed as mean ± SEM.
Source data are available online for this figure.

before even enough vector genome has reached the nucleus to initiate gene expression. Therefore, daily doses of intraperitoneal GCV and/or TMZ treatments started at day 13 post-tumor cell implantation and lasted for 6 days.

At day 27, post-cell implantation, *Luc* expression within the U87 tumors increased rapidly in groups of mice administered with the control non-targeted/AAVP-*Grp78-HSVtk* vector and GCV (Fig 4A). In contrast, although we observed an increase in *Luc* expression in

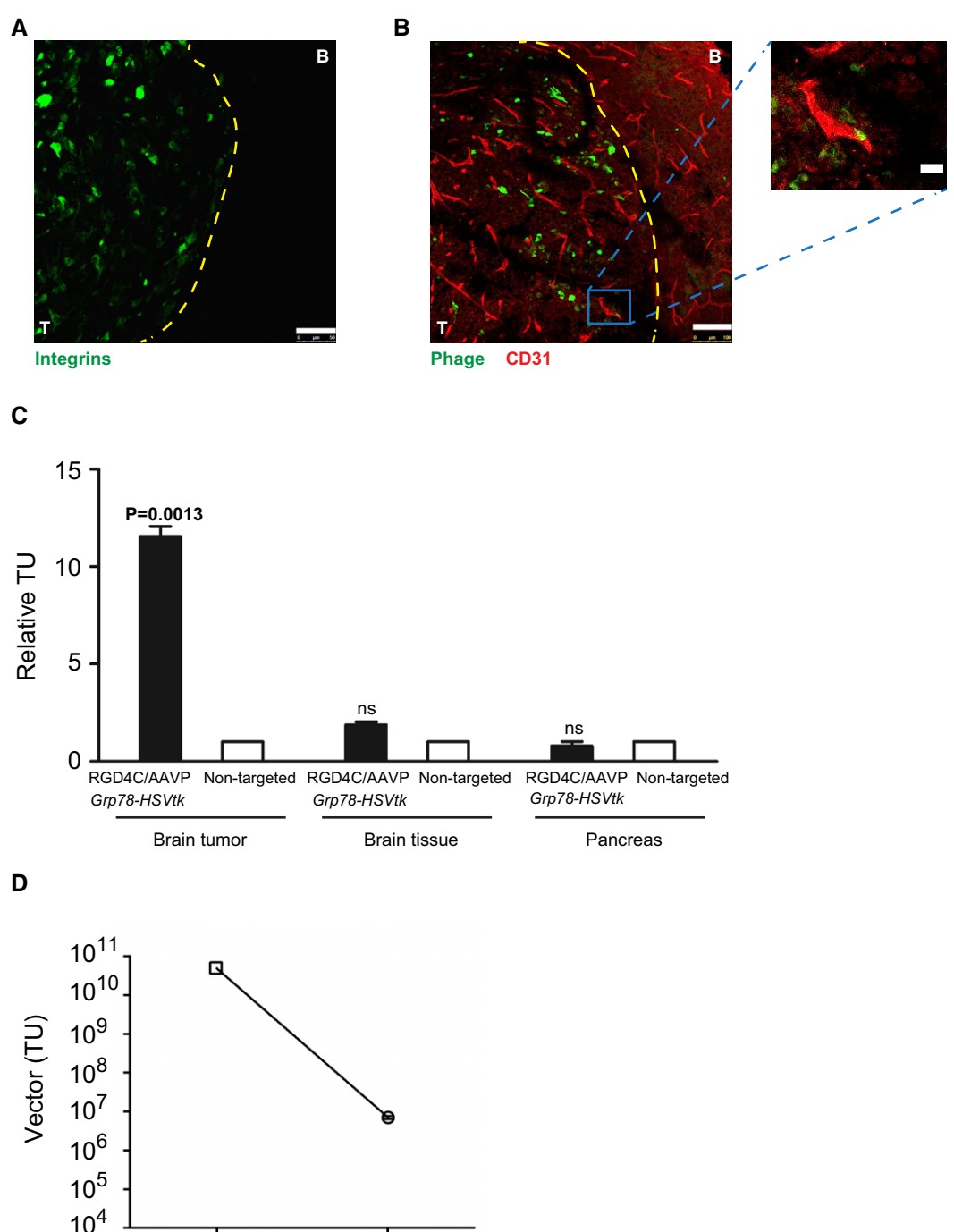

**Figure 3.  Systemic targeting of orthotopic glioblastoma with RGD4C/AAVP-*Grp78-HSVtk*.**

A   Immunofluorescent staining for $\alpha_v$ integrin expression in brain sections including intracranial tumor (T) and the surrounding healthy brain (B). Scale bar, 100 μm.

B   Co-staining of CD31 (red) and phage (green) in brain sections comprising tumor and healthy brain. Scale bar, 100 μm. A high-magnification view from the low-magnification insert of the tumor section is shown. Scale bar, 25 μm.

C   Quantification of the relative homing ability of RGD4C/AAVP-*Grp78-HSVtk* to brain tumors, brain tissue, and pancreas after intravenous administration. The data were normalized both to non-targeted vector and to tissue weight and then expressed as relative TU. Data shown are representative of two experiments, *n* = 5. One-way ANOVA with Tukey's multiple comparison test (GraphPad Prism 6) was used for data analysis.

D   Quantification of the vector dose reaching orthotopic GBM in mice upon intravenous administration of 5 × $10^{10}$ TU of RGD4C/AAVP vector. The vector amount, in tumors, was expressed as TU/g tumor tissue. Data shown are representative of two experiments, *n* = 5.

Data information: Data are expressed as mean ± SEM.
Source data are available online for this figure.

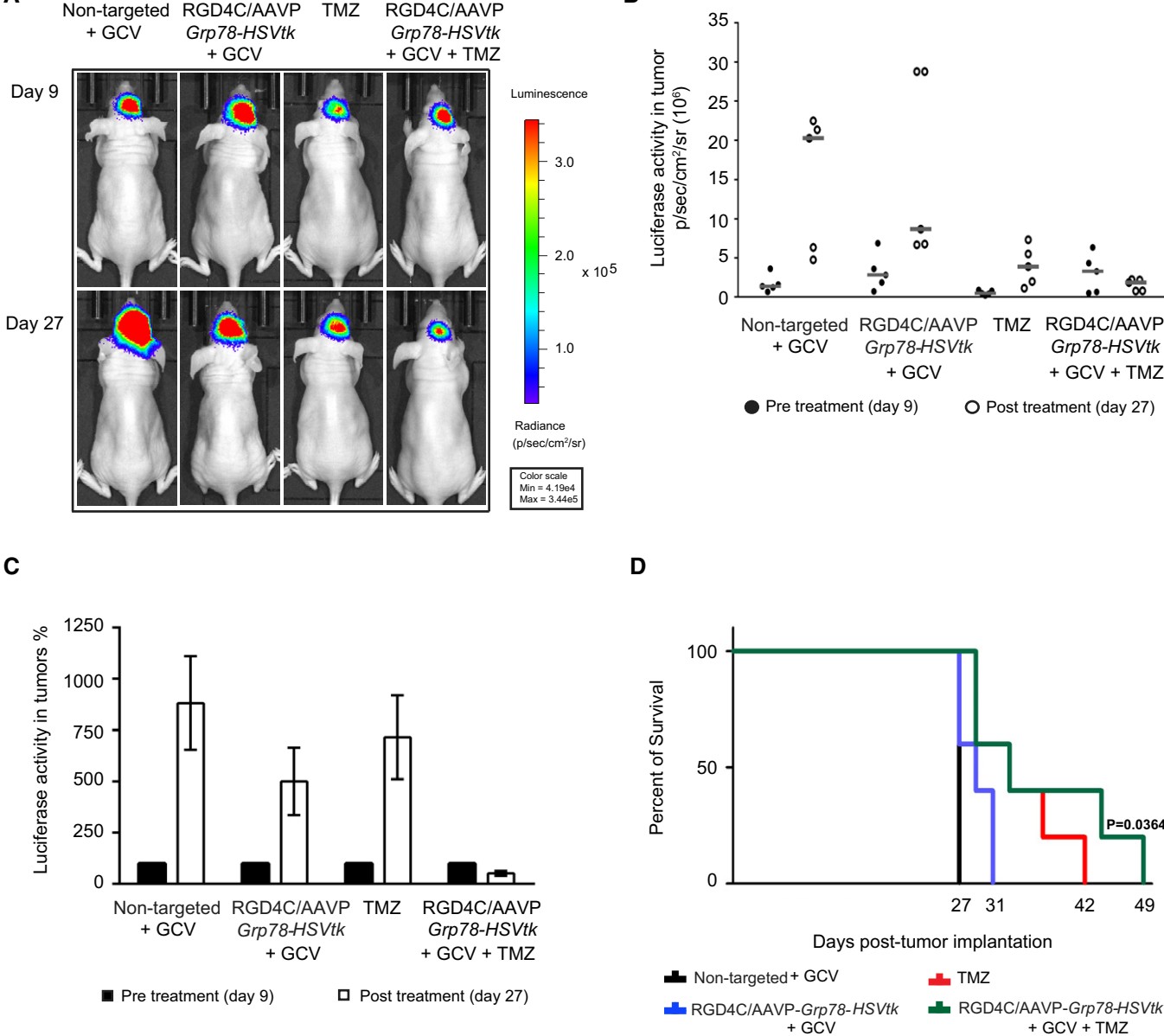

**Figure 4.   Therapeutic response of orthotopic U87 glioblastoma to combination of TMZ with RGD4C/AAVP-*Grp78-HSVtk* plus GCV.**

A   *In vivo* BLI of *Luc* expression for evaluation of tumor viability and tumor size of representative tumor-bearing mice from all experimental groups before initiation of therapy (day 9 post-intracranial cell implantation) and at the end of treatments (day 27 post-intracranial cell implantation).

B   Individual tumor *Luc* intensities before (filled circles) and after therapy (open circles), on days 9 and 27 post-cell implantation, respectively. Data shown are representative of two experiments, *n* = 5.

C   Average tumor *Luc* activity showing glioblastoma progression in each experimental group. Pre-treatment day 9 was set at 100%. Data shown are representative of two experiments, *n* = 5.

D   Survival benefit for tumor-bearing mice from all treatment groups. Data shown are representative of two experiments, *n* = 5. Statistical significance was determined by Kaplan–Meier method (Kaplan–Meier survival fractions). Log-rank (Mantel–Cox) test was used.

Data information: Data are expressed as mean ± SEM.
Source data are available online for this figure.

mice that received the RGD4C/AAVP-*Grp78-HSVtk* plus GCV, the *Luc* signals were reduced compared to the control non-targeted/ AAVP-*Grp78-HSVtk* plus GCV (Fig 4A–C), proving an inhibitory effect on tumor growth and tumor viability. Treatment with TMZ also resulted in tumor growth inhibition (Fig 4A–C). Remarkably, in sharp contrast, at day 27 post-tumor cell implantation, expression of the tumor *Luc* in mice receiving combination of RGD4C/AAVP-*Grp78-HSVtk*/GCV and TMZ was dramatically reduced than the

initial *Luc* signal recorded on day 9, showing not only a lack of tumor growth but a substantial reduction in tumor size and tumor viability (Fig 4A–C). Moreover, we evaluated efficacy on survival of mice with intracranial U87 tumors. All the treatments—RGD4C/ AAVP-*Grp78-HSVtk*/GCV, TMZ alone, and combination of RGD4C/ AAVP-*Grp78-HSVtk*/GCV plus TMZ—increased the survival of tumor-bearing mice (Fig 4D). Treatment with RGD4C/AAVP-*Grp78-HSVtk*/GCV produced better survival for tumor-bearing mice when compared to treatment with the non-targeted AAVP vector (Fig 4D). Importantly, tumor-bearing mice receiving combination treatment of RGD4C/AAVP-*Grp78-HSVtk*/GCV with TMZ showed the highest survival benefit as compared to TMZ or RGD4C/AAVP-*Grp78-HSVtk*/GCV alone.

Next, to check for post-treatment effects, tissues were snap frozen at day 27 post-tumor cell implantation and sectioned. The hematoxylin and eosin (H&E) staining of the brain sections revealed a clear damage to tumors following systemic gene therapy with RGD4C/AAVP-*Grp78-HSVtk* plus GCV compared to non-targeted/ AAVP-*Grp78-HSVtk*/GCV-treated tumors (Fig 5A). This anti-tumor effect was comparable to that observed in tumors treated with intraperitoneal TMZ. Remarkably, the H&E staining revealed extensive and greater tumor destruction from a single systemic dose of RGD4C/AAVP-*Grp78-HSVtk*/GCV plus TMZ treatment compared to vector or TMZ alone (Fig 5A). Tumors were also analyzed for cell proliferation by staining for the proliferation marker protein Ki67, a protein strictly associated with cell proliferation, and revealed that combination treatment of TMZ with RGD4C/AAVP-*Grp78-HSVtk*/ GCV generated the greatest reduction of cell proliferation compared to tumors from mice that received monotherapies of TMZ alone or RGD4C/AAVP-*Grp78-HSVtk*/GCV (Fig 5A). Moreover, immunofluorescent analysis of CD31 for blood vessel staining (Fig 5B) showed enlarged tumor blood vessels in mice that received the non-targeted/AAVP-*Grp78*-HSVtk/GCV, while the tumor blood vessel size was slightly reduced in mice treated with either RGD4C/AAVP-*Grp78-HSVtk*/GCV or TMZ alone. In contrast, mice that had received RGD4C/AAVP-*Grp78-HSVtk*/GCV plus TMZ showed normal blood vessel size in the remaining tumor lesion (Fig 5B). We also examined the tumors for apoptosis by evaluating expression of the caspase-3 which marks apoptotic cells, as the *HSVtk*/GCV is associated with apoptotic cell death. The data revealed that all treatments induced apoptosis within the tumors compared to tumors from mice that received the non-targeted control vector (Fig 5C). Interestingly, combination treatment with TMZ and RGD4C/AAVP-*Grp78-HSVtk*/ GCV induced the highest level of apoptosis in glioblastoma (Fig 5C). There was no weight loss noticed in the animals during the course of treatment (Fig EV2A), and any weight loss, detected by the end of the experiment, was solely related to the tumor burden.

To further confirm the vector safety, the toxicity of the RGD4C/ AAVP-*Grp78-HSVtk* was assessed in wild-type BALB/c mice administered intravenously with increasing doses $2.5 \times 10^9$ TU ($1 \times 10^{11}$ TU/kg), $1 \times 10^{10}$ TU ($5 \times 10^{11}$ TU/kg), or $5 \times 10^{10}$ TU ($2 \times 10^{12}$ TU/kg) of the RGD4C/AAVP-*Gp78-HSVtk* (Fig EV3). These doses were based on previous studies in mice and pet dogs (Paoloni *et al*, 2009), and on the clinical trial that performed intravenous injections of the M13 phage, parent of AAVP, to cancer patients (Krag *et al*, 2006).

Next, we did not observe any abnormal changes in the healthy tissues, no animal weight loss was noticed and the serum evaluation

of the lactate dehydrogenase (LDH), used as a biomarker for cellular cytotoxicity and cytolysis, remained at normal levels (Fig EV3).

## RGD4C/AAVP does transduce human primary GBM but not human normal cells

We sought to evaluate efficacy of vector in human primary GBM cells and performed several experiments by using two primary GBM cells named HSJD-GBM-001 (Olaciregui NG and Carcaboso AM, in preparation) and G26 (Pollard, 2013). GBM is a heterogeneous tumor with low survival that has been, at least partly, caused by glioma stem cells. These therapy-resistant GBM stem cell sub-populations are a sub-set of slow-cycling cells endowed with stem cell-like properties and able to resist the standard treatment and to sustain the relapse. Therefore, we first analyzed the HSJD-GBM-001 spheres and G26 cells for the presence of stem cell-like properties, since HSJD-GBM-001 are grown in serum-free medium, *in vitro*, to generate spheroids for *in vivo* implantation into animals. We found that HSJD-GBM-001 spheres express high levels of stem cell markers (65.7% Sox-$2^+$ cells, 62.7% of Nestin$^+$ cells, and 93.6% CD133$^+$ cells) proving that the HSJD-GBM-001 spheres contain high percentage of stem cells (Fig 6A). The G26 cells had 13.4% Sox-$2^+$ cells, 45.7% of Nestin$^+$ cells, and 43.5% CD133$^+$ cells. These data confirm the heterogeneity of HSJD-GBM-001 spheres and G26 cells.

Next, we analyzed these GBM primary cells for expression of the $\alpha_v\beta_3$ and $\alpha_v\beta_5$ integrin receptors. Immunofluorescence analyses showed that the HSJD-GBM-001 primary cells express high levels of the integrin subunits $\alpha_v$, $\beta_3$, and $\beta_5$ (Fig 6B); G26 cells also express these integrins but at lower levels than HSJD-GBM-001 spheres (Fig 6B). Then, we showed that the RGD4C/AAVP-*Luc* vector efficiently delivered gene expression to both HSJD-GBM-001 and G26 primary GBM, and gene delivery increased over time in a selective manner as no gene expression was detected in cells treated with the non-targeted vector, lacking the RGD4C ligand (Fig 6C). Gene delivery into HSJD-GBM-001 was higher than in G26 cells consistent with the higher integrin expression in HSJD-GBM-001 cells. For instance, at day 8, RGD4C/AAVP-*Luc* achieved Luc expression of ~ 570,000 RLU/µg protein in HSJD-GBM-001 cells compared to ~ 70,000 RLU/ µg protein in G26 cells (Fig 6C).

After demonstrating the potential of RGD4C/AAVP for gene delivery to human glioblastoma cells through the RGD4C binding to integrins, we sought to further assess the tumor cell selectivity of vector and its safety for human healthy cells. Thus, we assembled a panel of normal human primary cells from different histological origins, such as primary glial cells (astrocytes) and chondrocytes, and also included two types of human normal primary fibroblasts from skin and lung since $\alpha_v\beta_5$ integrin is also found expressed on normal fibroblasts. Firstly, we investigated expression of the integrins $\alpha_v\beta_3$ and $\alpha_v\beta_5$ receptors. While the astrocytes showed very low expression of $\alpha_v$ and $\beta_5$, with barely detectable levels of $\beta_3$, immunostaining of the chondrocytes did not show any expression of these integrins (Fig 7A). We also found that both primary fibroblasts have low levels of expression of $\alpha_v$ and $\beta_5$ subunits, with barely detectable $\beta_3$ integrin (Fig 7A). Importantly, this low integrin expression profile of normal cells did not translate into gene delivery by the RGD4C/AAVP-*Luc* vector alone or even in the presence of TMZ (Fig 7B), as the luminescent signals were not different from non-transduced cells at all time points. Consequently, unlike

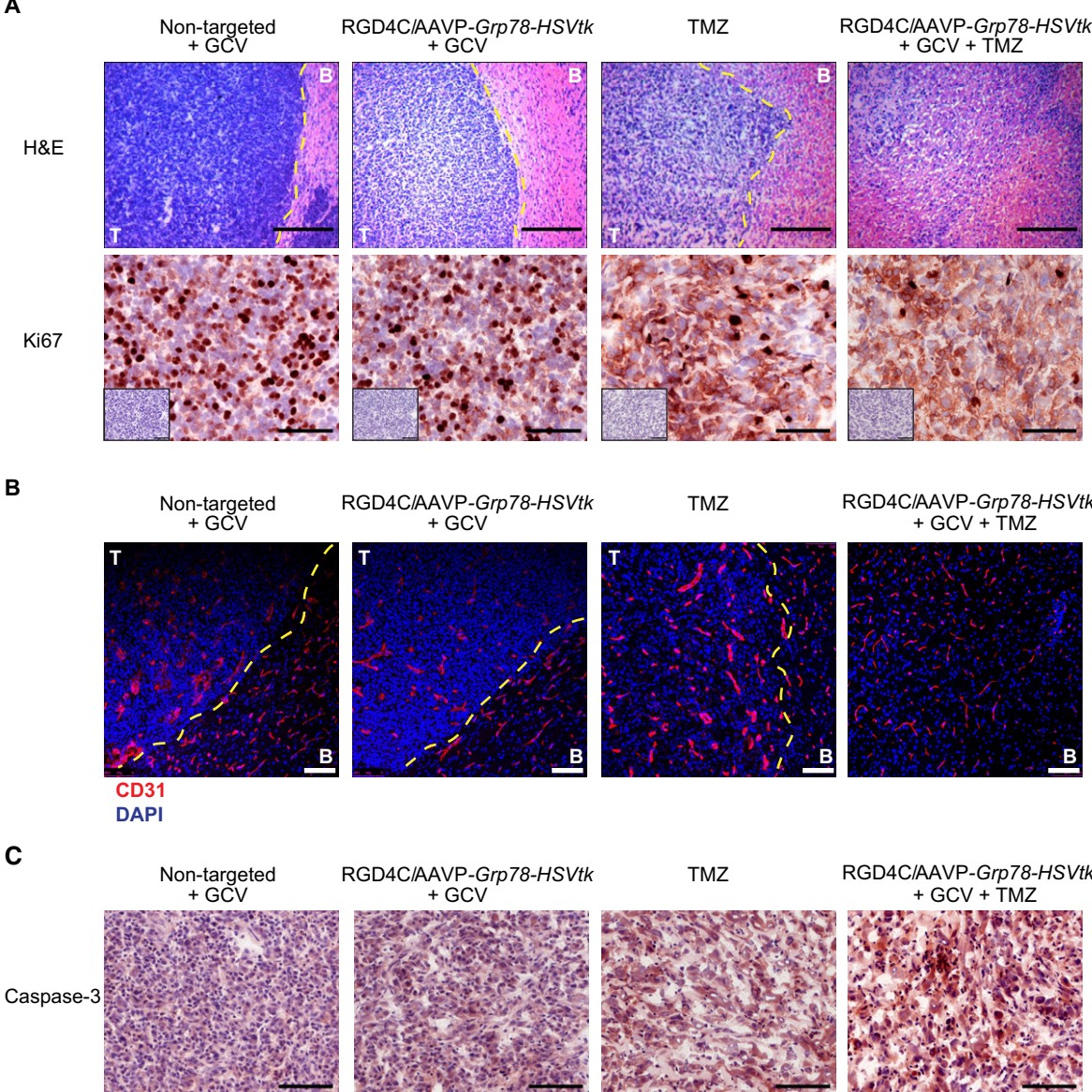

**Figure 5.  Histological analysis of intracranial U87-derived glioblastoma after therapy.**

A   Hematoxylin and eosin staining (H&E) and immunostaining for the proliferation marker protein Ki67 of representative tumor sections from all experimental groups, Scale bars, 200 μm for H&E and 100 μm for Ki67. The low-magnification inserts represent negative controls with the secondary antibody alone on serial sections used for both Ki67 and caspase-3 immunostainings.

B   Blood vessel staining with CD31 (red). DAPI staining of the sections is shown in blue. Scale bar, 200 μm.

C   Tumor immunostaining for apoptosis using an anti-caspase-3 antibody. Scale bar, 100 μm.

Data information: T, tumor; B, brain.

primary GBM, human normal cells show minimal or no expression of these integrin receptors, and this integrin expression profile did not permit any gene expression by the RGD4C/AAVP.

**Systemic targeting of primary GBM by RGD4C/AAVP-*Grp78-HSVtk* plus GCV inhibits tumor growth, and efficacy is enhanced by TMZ**

To rule out the possibility that the observed anti-tumor effects on orthotopic U87 GBM were specific to this GBM cell line, we analyzed therapy efficacy on intracranial primary GBM derived from the HSJD-GBM-001 spheres in nude mice, since they establish intracranial GBM in immunodeficient mice with 100% take rate (Olaciregui NG and Carcaboso AM, in preparation) and are heterogeneous with high levels of stem cell-like properties (Fig 6A). HSJD-GBM-001-*Luc* cells stably expressing the *Luc* reporter gene were grown in serum-free medium, *in vitro*, to generate spheres before implantation into the brain of nude mice. Next, tumors were established by intracranial GBM implantation as shown by the BLI of

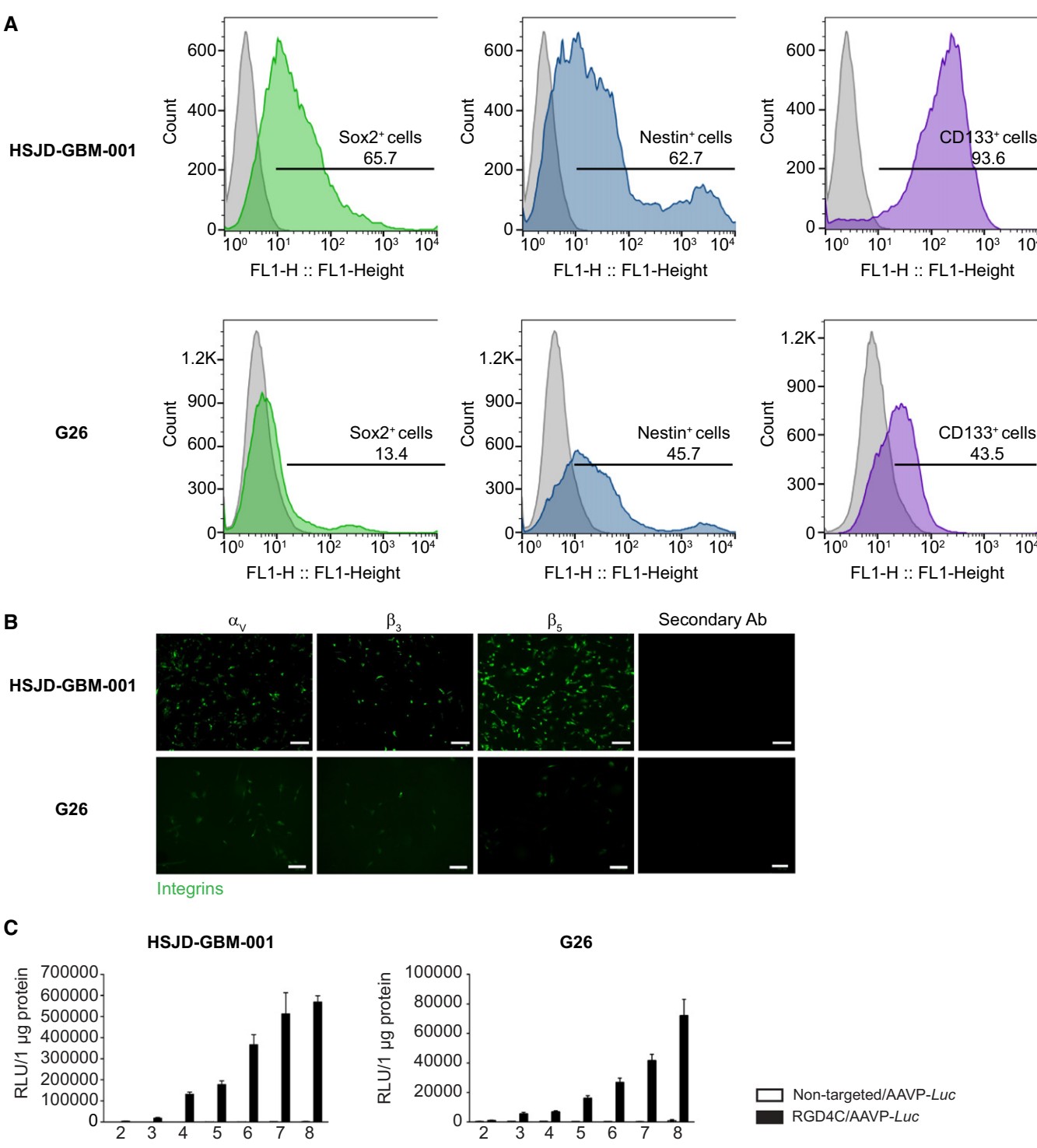

**Figure 6.  Characterization of the human primary GBM and then analysis of targeted gene delivery.**

A   Fluorescence-activated cell sorting (FACS) analysis with antibodies against Sox-2, Nestin, and CD133 stem cell markers.

B   Immunofluorescent staining for $\alpha_v$, $\beta_3$, or $\beta_5$ integrins in primary HSJD-GBM-001 and G26 GBM cells. Scale bars, 80 μm for HSJD-GBM-001 and 100 μm for G26.

C   Targeted transduction of HSJD-GBM-001 and G26 cells, over a time course, by RGD4C/AAVP-*Luc*. Non-targeted/AAVP-*Luc* was used as negative control. Results represent RLU/1 μg protein of triplicate wells. Data shown are representative of two experiments, *n* = 3. Data information: Data are expressed as mean ± SEM.

Source data are available online for this figure.

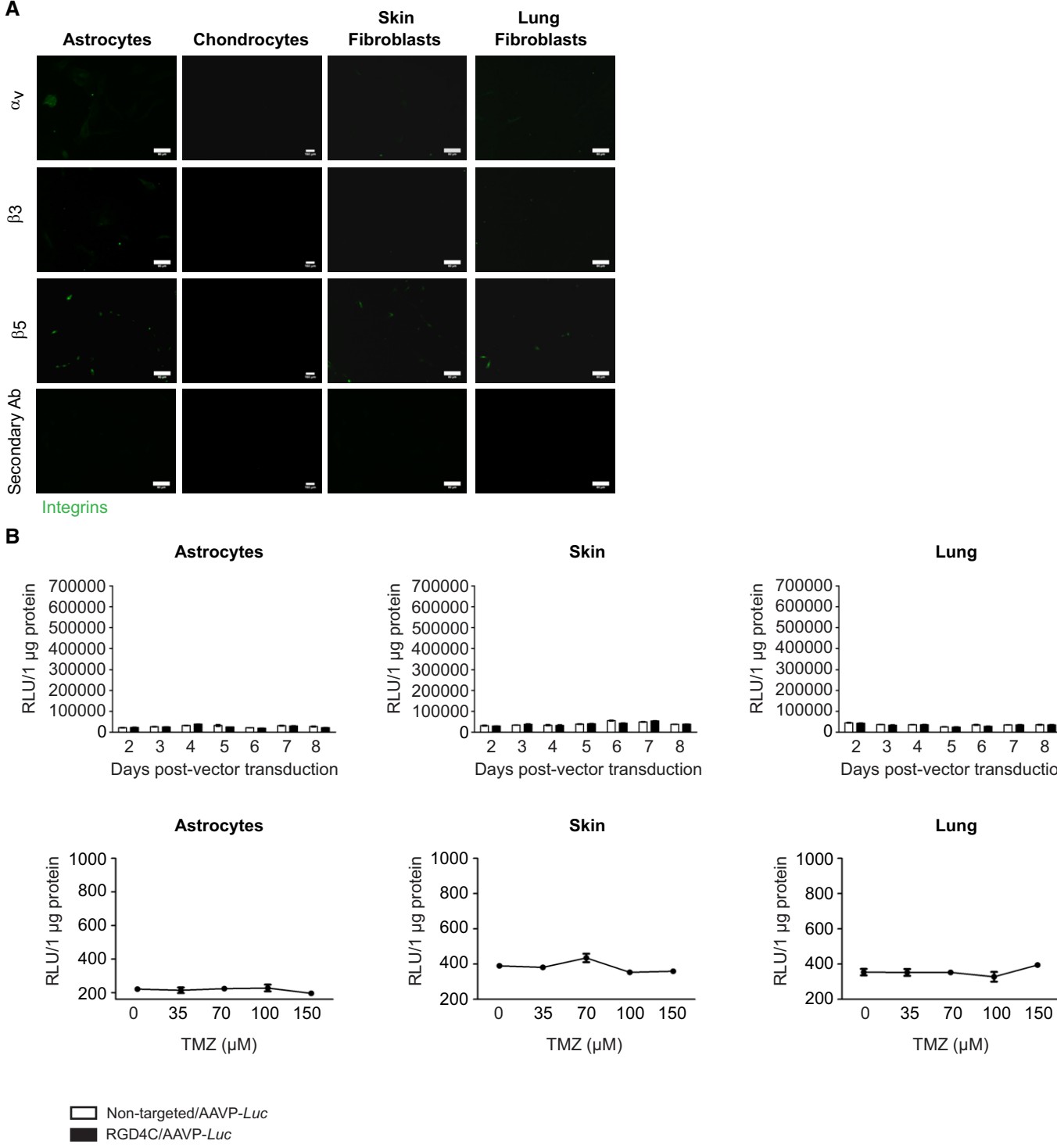

**Figure 7. Integrin expression in human primary normal cells and then analysis of targeted gene delivery.**

A  Immunofluorescent staining for $\alpha_v$, $\beta_3$, or $\beta_5$ integrins in human normal astrocytes, chondrocytes, skin, and lung fibroblasts. Scale bars, 160 μm for chondrocytes and 80 μm for all other cells.

B  Treatment of human primary normal cells with RGD4C/AAVP-*Luc* vector, alone or in the presence of TMZ. Non-targeted/AAVP-*Luc* was used as negative control. Results represent RLU/1 μg protein. Data shown are representative of two experiments, *n* = 3.

Data information: Data are expressed as mean ± SEM.
Source data are available online for this figure.

*Luc*; subsequently, the RGD4C/AAVP-*Grp78-HSVtk* or control vectors were intravenously injected into tumor-bearing nude mice. Consistent with U87 xenografts, daily treatment with GCV and TMZ was initiated at day 4 post-vector injection and continued for 6 days. At day 19 post-treatment initiation, BLI imaging of *Luc* showed that tumors from mice treated with the targeted RGD4C/AAVP-*Grp78-HSVtk* and GCV had lower Luc signals and were smaller compared to tumors treated with the control non-targeted vector and GCV (Fig 8A). Similar effects were also observed in mice receiving TMZ treatment (Fig 8A). Interestingly, barely detectable tumor Luc signals were observed in mice treated with combination of TMZ with RGD4C/AAVP-*Grp78-HSVtk* plus GCV compared to mice administered with RGD4C/AAVP-*Grp78-HSVtk* plus GCV or TMZ alone (Fig 8A). To support these imaging findings, we investigated whether the observed anti-tumor effects against primary GBM would translate into overall survival benefit, under the same experimental protocol above. The results showed a survival benefit in the mice treated with the RGD4C/AAVP-*Grp78-HSVtk* and GCV compared to the non-targeted vector/GCV-treated mice (Fig 8B, half the animals survived 54 days versus 47 days, respectively). Survival was also improved in TMZ-treated mice (Fig 8B, half the animals survived 66 days). Importantly, the best survival benefit was obtained in tumor-bearing mice treated with the combination therapy of TMZ with RGD4C/AAVP-*Grp78-HSVtk* and GCV, as compared to TMZ or vector alone (half the animals survived 72 days versus 66 and 54, respectively). Additionally, while there was no animal survival at day 54 post-cell implantation in mice treated with the control non-targeted vector, mice treated with combination RGD4C/AAVP-*Grp78-HSVtk*/GCV and TMZ could survive beyond day 90 post-cell implantation compared to days 66 and 73 for RGD4C/AAVP-*Grp78-HSVtk*/GCV and TMZ-treated mice, respectively (Fig 8B).

Moreover, we analyzed post-treatment effects and carried out detailed histopathological analysis of tumors recovered 12 days after therapy. Specifically, H&E staining revealed tumor destruction caused by the single systemic dose of RGD-4C/AAVP-*HSVtk* plus GCV, as well as by TMZ treatment, when compared to control tumors from mice injected with the non-targeted vector and GCV (Fig 8C). Interestingly, combination treatment further enhanced destruction of the central area of the tumor compared to each treatment alone (Fig 8C). Moreover, these findings were consistent with the staining for the proliferation marker protein Ki67, showing the lowest level of cell proliferation observed in tumors from mice treated with combination of TMZ and RGD4C/AAVP-*Grp78-HSVtk*/GCV, compared to each treatment alone (Fig 8C). Immunostaining for caspase-3 showed that apoptosis has become clearly visible in tumors from mice treated with combination of TMZ and RGD4C/AAVP-*Grp78-HSVtk* plus GCV (Fig 8C). Apoptosis was also induced in tumors from mice treated with either RGD4C/AAVP-*Grp78-HSVtk* plus GCV or TMZ. Finally, immunostaining of HSJD-GBM-001 tumors for the CD133[+] stem cell marker showed that tumors recovered from the control group are rich in stem cells proving the heterogeneity of the established intracranial tumors (Fig 8D). Interestingly, we detected clear reduction of cancer stem cells following treatment with RGD4C/AAVP-*Grp78-HSVtk* plus GCV, compared to TMZ treatment (Fig 8D). Notably, combination treatment with RGD4C/AAVP-*Grp78-HSVtk* plus GCV and TMZ resulted in marked suppression of the CD133[+] cells (Fig 8D).

There was no apparent toxicity as the mice did not lose significant weight during the course of the therapy (Fig EV2B), and the control organs removed from tumor-bearing mice treated by the same experimental protocol revealed no histopathologic abnormalities (Fig EV2C). Additionally, immunostaining against the RGD4C/AAVP-*Grp78-HSVtk* vector, using an antibody against the vector capsid, did not reveal any vector presence in the normal tissues (liver, kidney, heart, and lung) recovered after therapy (Fig EV2D). Importantly, tumors were stained positive for the vector upon administration of RGD4C/AAVP-*Grp78-HSVtk* or a combination of RGD4C/AAV-*Grp78-HSVtk* and TMZ (Fig EV2D).

### Targeted systemic gene therapy with RGD4C/AAVP-*Grp78-HSVtk* plus GCV inhibits tumor growth of orthotopic GBM in immunocompetent mice, and efficacy is boosted by TMZ

To rule out the possibility that the anti-tumor effects of this treatment regimen were specific to immunosuppressed mice, or either species or xenograft specific, we evaluated efficacy of the RGD4C/AAVP-*Grp78-HSVtk* on a standard mouse GBM model. We chose an isogenic tumor in which GL261 mouse glioblastoma cells are implanted intracranially to induce rapid growth of tumors in immunocompetent C57BL/6J mice. The GL261 tumor is the most recognized murine GBM model that has been extensively used for preclinical testing of therapeutic approaches for GBM (Oh *et al*, 2014). First, we confirmed that the GL261 cells express high levels of the integrin subunits $\alpha_v$, $\beta_3$, and $\beta_5$ receptors that mediate cell transduction by the RGD4C/AAVP vector (Fig 9A). Next, to establish orthotopic GBM for therapy experiments, we used the GL261-*Luc* labeled with the *Luc* gene to allow bioluminescent imaging (BLI). Following brain tumor detection by BLI, tumor-bearing mice were injected intravenously with targeted RGD4C/AAVP-*Grp78-HSVtk* or control non-targeted vectors ($5 \times 10^{10}$ TU/mouse). Daily treatment with GCV and TMZ was initiated at day 4 post-vector injection and lasted for 5 days. Moreover, this treatment regimen was repeated three times to complete a total of three serial vector administrations, as we previously reported efficacy of repeated administrations of RGD4C/AAVP in immunocompetent BALB/c mice and immunocompetent pet dogs with natural tumors (Hajitou *et al*, 2006; Paoloni *et al*, 2009). At day 34 post-treatment initiation, BLI imaging revealed that the group of tumor-bearing mice treated with the targeted RGD4C/AAVP-*Grp78-HSVtk* plus GCV had smaller tumors with lower viability as shown by the total *Luc* signals in tumors, compared to the large size and highly viable tumors from mice receiving the control non-targeted vector and GCV (Fig 9B and C). Comparable effects were also observed in mice treated with TMZ (Fig 9B and C). Remarkably, no tumor *Luc* signals were observed in mice treated with combination of TMZ with RGD4C/AAVP-*Grp78-HSVtk* plus GCV, indicating striking tumor suppression by this combination treatment (Fig 9B and C).

Moreover, we investigated whether the repeated treatment regimen resulted in extended overall survival. There was a survival benefit following administrations of RGD4C/AAVP-*Grp78-HSVtk* and GCV compared to the non-targeted vector/GCV; survival was also improved in TMZ-treated group (Fig 9D). For instance, while there was no animal survival at day 38 post-tumor cell implantation in the control group, mice treated with RGD4C/AAVP-*Grp78-HSVtk*/

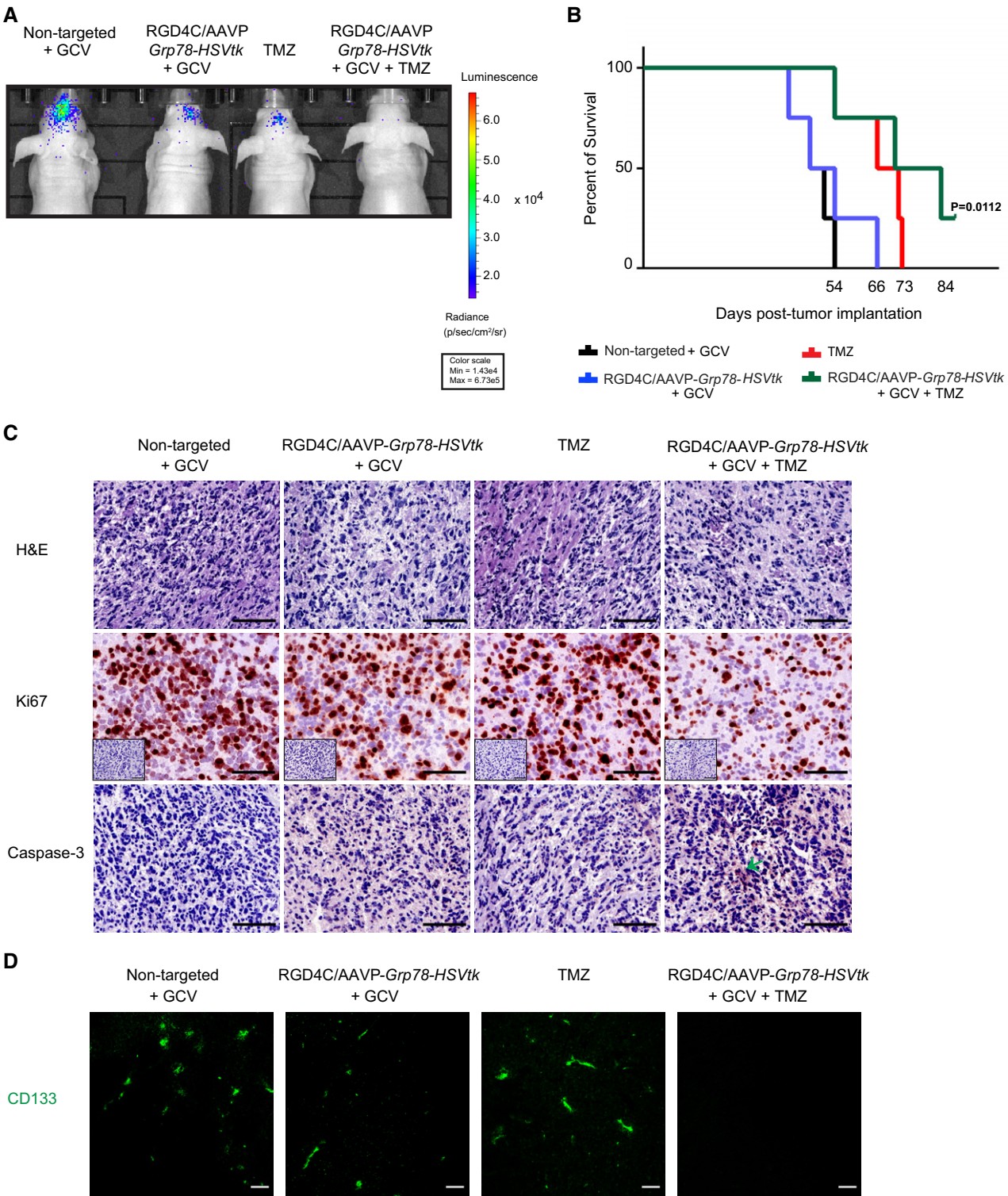

**Figure 8. Targeted systemic gene therapy against orthotopic primary HSJD-GBM-001 with RGD4C/AAVP-*Grp78-HSVtk* plus GCV and combination with TMZ.**

A BLI of luciferase of representative tumor-bearing mice from all experimental groups at day 19 post-therapy initiation.

B Survival benefit for tumor-bearing mice from all treatment groups. Data shown are representative of two experiments, *n* = 6. Statistical significance was determined by Kaplan–Meier method (Kaplan–Meier survival fractions). Log-rank (Mantel–Cox) test was used.

C H&E staining, Ki67 immunostaining, and evaluation of apoptosis in tumors by using an anti-caspase-3 antibody staining of representative tumor sections from all experimental groups. The low-magnification inserts represent negative controls with the secondary antibody alone on serial sections used for both Ki67 and caspase-3 immunostainings. Arrow points to apoptotic cells. Scale bars, 100 μm.

D Immunostaining of tumors with an antibody against CD133. Scale bars, 100 μm.

 

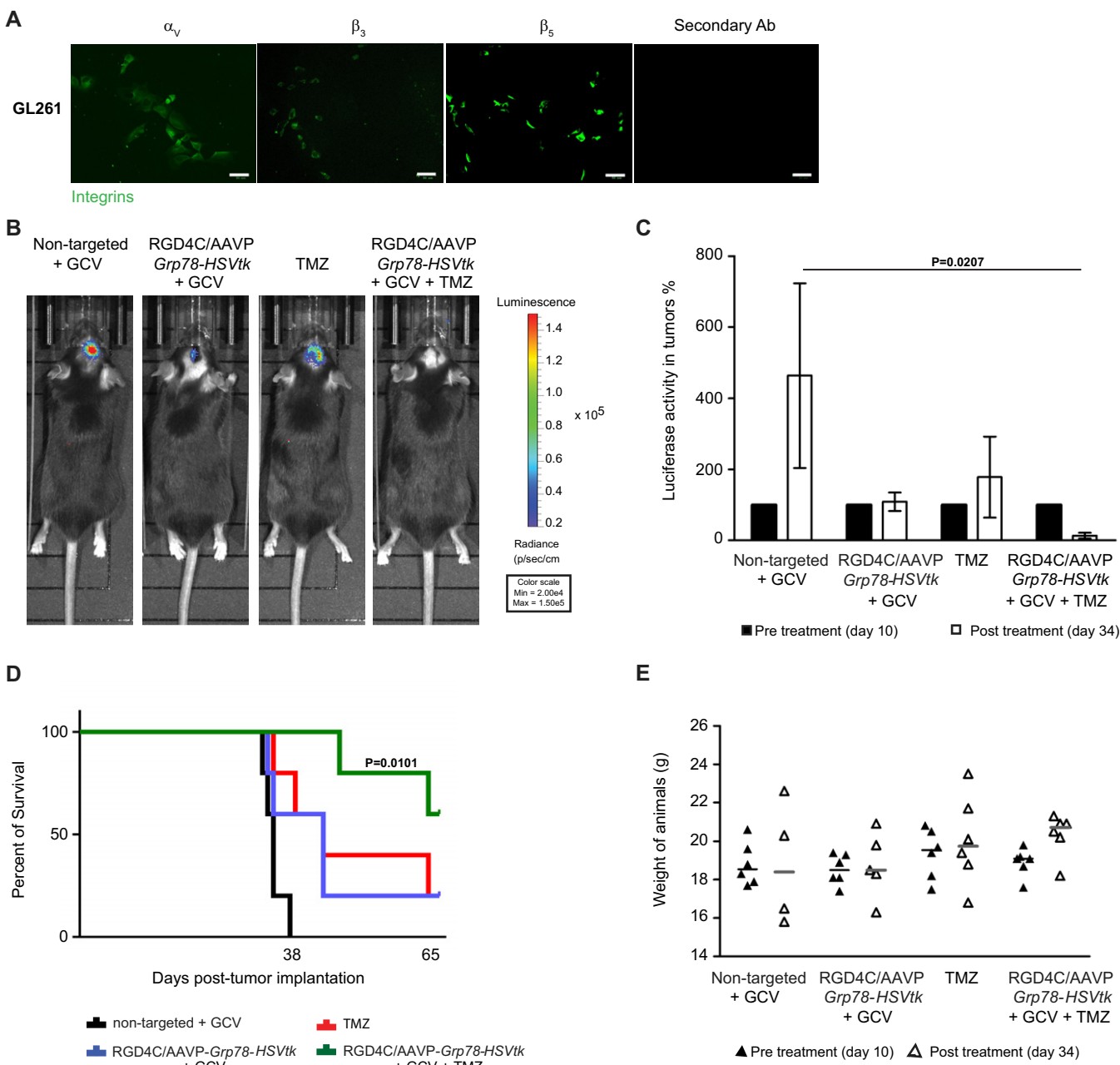

**Figure 9. RGD4C/AAVP-*Grp78-HSVtk* plus GCV gene therapy and combination with TMZ against orthotopic murine GL261 GBM.**

A   Immunostaining analysis for $\alpha_v$, $\beta_3$, or $\beta_5$ integrins in GL261 cells. Scale bar, 50 μm.

B   BLI of *Luc* of representative tumor-bearing mice from all experimental groups at day 34 post-treatment.

C   Total tumor *Luc* intensities in each experimental group before treatment, on day 10 post-cell implantation, and after therapy, on day 34 post-cell implantation. Data shown are representative of two experiments, $n = 6$. Data analysis was done by two-way ANOVA and Bonferroni's multiple comparison test.

D   Survival benefit for tumor-bearing mice from all treatment groups. Data shown are representative of two experiments, $n = 6$. Statistical significance was determined by Kaplan–Meier method (Kaplan–Meier survival fractions). Log-rank (Mantel–Cox) test was used.

E   Weights of C57BL/6J mice bearing intracranial GL261 tumors, from all experimental groups, before and after treatment. Data shown are representative of two experiments, $n = 6$.

Data information: Data are expressed as mean ± SEM.
Source data are available online for this figure.

GCV or TMZ survived beyond day 62 (Fig 9D). Remarkably, 80% of mice treated with combination RGD4C/AAVP-*Grp78-HSVtk*/GCV and TMZ could survive past day 62 post-cell implantation (Fig 9D).

There was no noticeable toxicity as there was no weight loss noted in the animals during the course of therapy (Fig 9E), and any weight loss detected by the end of the experiment was exclusively

associated with the intracranial tumor burden. Moreover, H&E staining of healthy tissues did not reveal any tissue damage or abnormalities following repeated treatment with the combination regimen (Fig EV4).

## Discussion

In this study, we combined the chemotherapeutic drug TMZ with RGD4C/AAVP-*Grp78*-mediated systemic gene therapy, delivered intravenously in mice, against intracranial human models of glioblastoma. We showed that RGD4C/AAVP delivers gene expression *in vitro* in a diverse panel of human GBM cell lines, primary GBM and GBM stem cells in a targeted and efficient manner, mediated through binding of the RGD4C to $\alpha_v\beta_3$ integrin, as all tumor cells and GBM stem cells tested in this study express this integrin and no gene delivery was obtained with vector lacking the RGD4C ligand. The $\alpha_v$, $\beta_3$, and $\beta_5$ subunits form integrin heterodimers such as $\alpha_v\beta_3$ that can bind the RGD4C ligand and internalize the RGD4C/AAVP, and the $\alpha_v\beta_5$ heterodimer that can also bind RGD4C, but to a lesser extent than $\alpha_v\beta_3$. Targeting $\alpha_v\beta_3$ integrin in human GBM with RGD4C was performed in patients (Reardon *et al*, 2008). Previous studies reported that $\alpha_v\beta_3$ integrin can be expressed in up to 55% of human glioblastoma, depending on the tumor grade (Schnell *et al*, 2008; Schittenhelm *et al*, 2013). The $\alpha_v\beta_3$ expression was significantly higher in human GBMs than in low-grade gliomas and was seen on both activated endothelial cells and glial tumor cells existing within glioblastomas (Schnell *et al*, 2008; Schittenhelm *et al*, 2013). In contrast, $\alpha_v\beta_3$ expression is barely detectable on normal human endothelial cells (Koch, 2000; Pap *et al*, 2000; Van De Wiele *et al*, 2002; Kumar, 2003; Gamble *et al*, 2010) and is also not expressed on glia or neurons in normal adult cortex and cerebral white matter (Gladson & Cheresh, 1991). Moreover, in the present study, we confirmed that the minimal expression of these integrin heterodimers in normal human primary cells including glial cells did not lead to gene delivery by the targeted RGD4C/AAVP-*Grp78* vector.

Then, we found that TMZ stimulates gene expression from the *Grp78* promoter, but not from the *CMV* promoter, in GBM cells by activating the UPR stress pathway that ultimately results in the stimulation of *Grp78* promoter. TMZ-induced expression and activation of the endogenous *GRP78* promoter have been reported in human glioblastoma (Pyrko *et al*, 2007; Virrey *et al*, 2008), and induction of *Grp78* has been reported in various cancer therapies, including chemotherapy (Li & Lee, 2006; Lee, 2007; Pyrko *et al*, 2007; Virrey *et al*, 2008). *Grp78* is a stress-inducible gene that encodes for a potent anti-apoptotic protein, which plays a critical role in tumor survival and resistance to therapy (Yu *et al*, 1999; Lee, 2001; Li & Lee, 2006; Daneshmand *et al*, 2007; Nagelkerke *et al*, 2014). In addition to TMZ, other therapeutic approaches can also be used to induce the *Grp78* promoter, e.g., radiation therapy (Sun *et al*, 2017) and other chemotherapeutic drugs used for brain tumor treatment such as cisplatin (Mandic *et al*, 2003). Moreover, we demonstrated that safe anti-cancer agents such as curcumin from natural sources, known for its ability to cross the BBB (Klinger & Mittal, 2016), have the potential to induce the RGD4C/AAVP-*Grp78* vector promoter in human primary gliomas, consistent with previous studies reporting induction of the *Grp78* in cancer cells, by curcumin, through the UPR pathway (Kim *et al*, 2016).

In the present study, combination of TMZ with vector carrying the *HSVtk* gene increased the cell death of GBM cell lines *in vitro* but the effect of combining TMZ with RGD4C/AAVP-*Grp78* was more pronounced than of TMZ combined with RGD4C/AAVP-*CMV*, which could be explained by TMZ induction of gene expression from the *Grp78* promoter. The difference between RGD4C/AAVP-*Grp78*/TMZ and RGD4C/AAVP-*CMV*/TMZ combinations was less pronounced in cell killing than in *Luc* gene expression analyses. However, it is important to note that the *HSVtk/GCV* approach induces a bystander effect (Trepel *et al*, 2009), which can result in increased cell killing by the RGD4C/AAVP-*CMV*. Thus, the cell killing results are partially proportional to the transduction efficiency.

Some of our results merit further discussion. Although the SNB19 cells exhibited a slower activation of the *Grp78* promoter by TMZ, these cells showed the best response to TMZ addition as compared to LN229 and U87 cell lines. However, these cells also showed the most response of *Grp78* promoter to addition of GCV. Therefore, it is possible that the highest response of SNB19 cells to TMZ plus GCV combination might, at least in part, be associated with the elevated response of *Grp78* promoter, in these cells, to GCV treatment. We previously reported that *HSVtk*/GCV therapy upregulates *Grp78* and transgene expression in glioma cells via the conserved UPR signaling cascade (Kia *et al*, 2012). On the other hand, cell death was measured at day 4 post-GCV addition, at which time point the *HSVtk*/GCV bystander effect might take over resulting in increased cell killing by the RGD4C/AAVP-*Grp78-HSVtk* and GCV. Thus, the cell killing data should not be completely proportional to the levels of induction of *HSVtk* expression. We have also confirmed that the cells express connexin-26 (Fig EV5); connexins are proteins associated with the bystander effect as they compose the channels of the GJIC through which toxic phosphorylated GCV is exchanged between cells.

*In vivo*, we found that systemic combination of TMZ with a single intravenous dose of RGD4C/AAVP-*Grp78-HSVtk* and GCV yielded synergistic and more pronounced anti-tumor effect than gene therapy or chemotherapy alone, both against orthotopic U87-derived xenografts and human primary GBM. Various factors could contribute to this striking anti-tumor effect. Firstly, an increase in TMZ-induced *HSVtk* expression from RGD4C/AAVP-*Grp78-HSVtk* by TMZ could reach high therapeutic *HSVtk* levels and subsequently increase the effect of targeted gene therapy. Secondly, synergy between *HSVtk* and TMZ against human glioblastoma has been reported (Rainov *et al*, 2001). Finally, a likely advantage of using RGD4C ligand is that it can target both the accessible tumor vasculature and the tumor cells giving the vector a dual mechanism for inducing tumor death. Previous studies reported that *Grp78* was induced by TMZ in both tumor cells and associated vasculature in GBM (Pyrko *et al*, 2007; Virrey *et al*, 2008). Our findings demonstrated that RGD4C/AAVP-*Grp78* is localized in both compartments of intracranial human GBM in mice. A notable finding from the *in vivo* studies is the ability of RGD4C/AAVP-*Grp78-HSVtk* vector plus GCV to produce a destruction of the GBM stem cells *in vivo* within the tumors and its enhancement in the presence of TMZ. These data are supported by our *in vitro* studies demonstrating efficient transduction of primary GBM stem cells by the RGD4C/AAVP vector. These findings are encouraging since GBM are heterogeneous tumors containing a sub-population of cells with stem cell-like properties that are resistant to standard treatment. Moreover,

developing a selective viral vector for GBM stem cells has been challenging (Dey *et al*, 2011); the RGD4C/AAVP vector holds promise for targeted delivery of therapeutic nucleic acids to GBM stem cells.

Immunostaining experiments revealed that RGD4C/AAVP-*Grp78*-HSVtk accumulates in intracranial tumors, derived both from the U87 and primary GBM cells, following intravenous administration. In contrast, no presence of vector was observed in the surrounding healthy brain and other vital normal tissues, consistent with previous studies. We and others previously reported the ability of RGD4C/AAVP to home selectively to tumors in preclinical models in mice and rats as well as in natural tumors in large animal models, while sparing the normal tissues, and subsequently deliver gene expression to tumors selectively without any detectable gene expression in the healthy tissues (Hajitou *et al*, 2006, 2007, 2008; Paoloni *et al*, 2009; Tandle *et al*, 2009; Kia *et al*, 2012; Przystal *et al*, 2013; Yuan *et al*, 2013; Dobroff *et al*, 2016; Smith *et al*, 2016). Herein, our findings revealed that the selective tumor homing is consistent with $\alpha_v$ integrin expression in the tumors but not in healthy brain. Some of these results merit further discussion. The ability of phage to cross the BBB was described as early as 1943 by Dubos *et al* (Dubos *et al*, 1943) in an investigation to treat meningitis in mice with intraperitoneal phage mixtures. In 2002, Frenkel and Solomon (Frenkel & Solomon, 2002) reported that intranasal administration in mice resulted in accumulation of M13 phage, parent of AAVP, in the brain. Moreover, an elegant study showed the ability of a glioma-binding M13 phage to cross the BBB in intracranial human gliomas in mice and achieve selective tumor homing upon intravenous administration (Ho *et al*, 2004, 2010). There is no clear mechanism for RGD4C/integrin-guided AAVP penetration into brain tumors. Our hypothesis is that the RGD4C/AAVP diffusion within the tumors, dual targeting of both the tumor vasculature and tumor cells and induction of *Grp78* promoter by TMZ could account for the therapeutic potential of RGD4C/AAVP-*Grp78*-HSVtk/GCV and TMZ combination against GBM.

In conclusion, these studies might be able to alter clinical use of TMZ chemotherapy of GBM but may also influence intravenous targeted gene therapy with RGD4C/AAVP-*Grp78* against GBM. Combinatorial treatment regimens more often constitute a successful approach to overcome resistance to therapies, which is a hallmark of GBM. Beyond that, potential synergistic interactions between the combined TMZ and RGD4C/AAVP-*Grp78*-HSVtk/GCV should permit possible dose reductions of TMZ, both to a less toxic and to a less costly degree. Cells transduced by RGD4C/AAVP-*Grp78*-HSVtk unintentionally enhance their own death upon TMZ dose/duration regimen treatment by activating the UPR stress pathway and ultimately *HSVtk* expression from the RGD4C/AAVP-*Grp78*-HSVtk. Phage-guided anti-cancer therapy can enter clinical trials in cancer patients as bacteriophages have long been safely administered to humans, from their antibacterial use during the pre-antibiotic era (Asavarut & Hajitou, 2014) to the Food and Drug Administration approval of certain phage preparations as antibacterial food additives (Lang, 2006). In this study, we also confirmed the anti-tumor effect of this treatment regimen against GBM established in immunocompetent mice by applying three serial vector administrations that were safely accomplished without any unwanted adverse reactions. The M13 phage, parent of RGD4C/AAVP-*Grp78* vector, was well reported for its immunogenicity (Trepel *et al*, 2001). Importantly however, we and collaborators have established that repeated administrations of RGD4C/

AAVP-*HSVtk* proved safe and efficient to inhibit tumor growth in wild-type animals and subsequently improve their survival (Hajitou *et al*, 2006; Paoloni *et al*, 2009). Moreover, clinical trials in cancer patients showed that serial intravenous phage library administration, based on the M13 phage, can be accomplished without major untoward clinical effects (Arap *et al*, 2002; Krag *et al*, 2006) despite the presence of anti-M13 phage IgGs. Phage-based particles are known to be immunogenic, but this feature can be modulated through targeting itself (Trepel *et al*, 2001).

# Materials and Methods

### Cell culture

LN229, U87, and SNB19 cell lines were purchased from American Type Culture Collection (ATCC). GL261 cells were obtained from The Leibniz Institute DSMZ, Germany. Primary human skin and lung fibroblasts were a gift from Dr. David Abraham, Royal Free Hospital, United Kingdom. Primary human chondrocytes were a gift from Dr. Peraphan Pothacharoen, Chiang Mai University, Thailand, and primary astrocytes were obtained from the ATCC. All these cells were maintained in Dulbecco's modified Eagle's medium (DMEM) supplemented with 10% fetal bovine serum (FBS), except astrocytes that were grown in the Astrocyte medium supplemented with Astrocyte supplements.

Primary human GBM HSJD-GBM-001 and primary pediatric human glioma were established at the Hospital Sant Joan de Deu, Barcelona, Spain, and were grown in tumor stem medium (TSM; 50% Neurobasal-A Medium (1×), 50% DMEM/F12 (1×), 1% HEPES buffer solution (1 M), 1% sodium pyruvate MEM (100 mM), 1% MEM Non-Essential Amino Acids solution 10 mM (100×), 1% GlutaMAX supplement, 1% Antibiotic-Antimycotic (100×), and 10% FBS).

Primary Human GBM G26 (provided by Pr. Steven Pollard from the Edinburgh Brain Cancer, University of Edinburgh, Scotland) was maintained in serum-free cultures on laminin, using neural stem (NS) cell media supplemented with EGF and FGF-2 to final concentration of 10 ng/ml as reported (Pollard, 2013).

GCV was used at 10 μmol/l, while TMZ was applied at various concentrations depending on the experiment and the cell type, as indicated in the manuscript, and renewed daily. Curcumin was added to cells at day 3 post-vector transduction and gene expression measured at day 6.

Expression of the luciferase, *Luc*, reporter gene *in vitro* was determined by a luciferase enzymatic assay using The Promega Steady-glo® Luciferase Assay Kit according to the manufacturer's instructions and then quantified with a Promega Glomax microplate reader.

Transduction of cells by AAVP vectors was performed in serum-free media as reported (Hajitou *et al*, 2007).

### Production, purification, and titration of AAVP vectors

AAVP vectors were generated as previously reported (Hajitou *et al*, 2006; Kia *et al*, 2012) by inserting a recombinant rAAV2 genome containing the reporter or therapeutic genes into the fUSE5 plasmid derived from the filamentous fd-tet bacteriophage. AAVP viral

particles were produced and purified from the culture supernatant of *Escherichia coli* K91kan host bacteria, then sterile-filtered through 0.45-μM filters (Hajitou *et al*, 2006). The AAVP titer was calculated by infecting K91 host bacteria and expressed as bacterial transducing units (TU/μl) as reported (Hajitou *et al*, 2006, 2007).

### Integrin staining in cells

Tumor and normal cells were stained for $\alpha_v$, $\beta_3$, and $\beta_5$ integrins on poly-D-lysine-coated coverslips in 12-well plates as previously reported (Stoneham *et al*, 2012). Cells were viewed, and images were taken using a Nikon Eclipse TE2000-S fluorescence microscope.

### Western blot

Whole-cell lysates were prepared in radioimmunoprecipitation assay buffer (RIPA) and subjected to immunoblot. We used goat anti-Grp78 (C-20, 1:400), mouse anti-GAPDH (6C5, 1:1,000), mouse anti-ATF6 (IMG-273, 5 μg/ml) from Imgenex, USA, and rabbit-anti phosphor-eIF2α (1:1,000) from Cell Signaling. We also used a mouse anti-connexin 26 antibody (1:1,000). Each immunoblot was done three times, quantified by ImageJ software, and normalized to GAPDH.

### *XBP1* splicing measurement

To detect unspliced and spliced forms of the *XBP1*, semi-quantitative RT–PCR was performed as we previously described (Kia *et al*, 2012) by using the forward TTACGAGAGAAAACTCATGGCC and reverse GGGTCCAAGTTGTCCAGAATGC primers.

### MitoSOX™ Red staining for Fluorescence-Activated Cell Sorting (FACS) and Sulforhodamine B (SRB) assay

MitoSOX™ Red reagent is a live-cell dye that is rapidly and selectively targeted to the mitochondria to stain the ROS/superoxide. MitoSOX™ Red staining was performed according to the manufacturer's instructions. The SRB assay was carried out as previously reported (Vichai & Kirtikara, 2006).

### Animal models and anti-tumor therapy

Human intracranial tumors were established in immunodeficient mice by using the U87 cell line or HSJD-GBM-001 spheres, while the GL261 cells were implanted intracranial into immunocompetent C57BL/6J mice to generate murine GBM. These cells were first labeled with the *Luc* reporter gene by using a lentiviral vector (System Biosciences) to generate the U87-*Luc*, HSJD-GBM-001-*Luc*, and GL261-*Luc* cells. Then, a total of $2.5 \times 10^5$ U87-*Luc*, $5 \times 10^5$ HSJD-GBM-001-*Luc*, or $1.0 \times 10^5$ GL261-*Luc* cells were implanted into the brain of mice. Tumor-bearing mice were intravenously administered through the tail vein with targeted or control non-targeted vectors carrying the *HSVtk* at a dose of $5 \times 10^{10}$ TU vector/mouse (Hajitou *et al*, 2006, 2007).

In therapy experiments, GCV (70 mg/kg/day) and TMZ (30 mg/kg/day) were administered intraperitoneally. Tumor growth was monitored two to three times a week by BLI of *Luc* using the *In Vivo* Imaging System (IVIS 100; Caliper Life Sciences). A region of interest was defined manually over the tumors for

measuring signal intensities recorded as total photon counts per second per $cm^2$ (photons/sec/$cm^2$/sr; Hajitou *et al*, 2006, 2007). At the end of therapy, mice were killed by terminal perfusion through the heart, and then, the tumors and normal tissues were harvested.

In tumor homing experiments, mice were perfused at 18 h post-vector administration, and then, the tumors and control organs were removed and weighted. Tissues were then ground with a glass Dounce homogenizer, suspended in 1 ml of DMEM containing proteinase inhibitors (1 mM phenylmethylsulfonyl fluoride (PMSF), 20 μg/ml aprotinin, and 1 μg/ml leupeptin), vortexed, and washed three times with DMEM. To quantify vectors, tissue homogenates were incubated with 1 ml of host bacteria (log phase *E. coli* K91kan; $OD600 \approx 2$). Aliquots of the bacterial culture were plated onto Luria–Bertani agar plates containing 40 μg/ml tetracycline and 100 μg/ml of kanamycin. Plates were incubated overnight at 37°C, followed by colony counting, to determine the amount of functional RGD4C/AAVP as TU and subsequently evaluate the vector dose that reaches the whole tumor versus the healthy tissues upon intravenous administration of $5 \times 10^{10}$ TU of RGD4C/AAVP particles. Vector accumulation in tumors was expressed as TU/g of tumor tissue.

In the toxicity study in wild-type female BALB/c mice, quantitative evaluation of the LDH was done by using the CytoTox 96® colorimetric Cytotoxicity Assay (Promega).

We have used 5–6 mice per group and repeated the experiments twice. Mice nude female and immunocompetent C57BL/6J or BALB/c female mice, adult 5–7 weeks, were purchased from Charles River, United Kingdom. Athymic nude mice are more prone to infection by opportunistic pathogens and were thus housed behind appropriate barrier housing; irradiated diet and bedding were also provided to reduce the risk of infection. Experiments involving living mice were carried out according to the Institutional and Home Office Guidelines, and under a granted Home Office-issued project license. The project license was first reviewed and approved by the Animal Welfare and Ethical Review Body (AWERB committee) at Imperial College London, before its final review and approval by the Home Office.

### Immunohistochemistry

Tumor vascularization, phage immunodetection, and integrin expression were assessed on frozen sections by using antibodies against CD31 (vascular marker), phage, and $\alpha_v$ integrin. Frozen sections (8 μm) were washed with 1×PBS (phosphate buffer saline) and 0.3% Triton X-100 (PBS-Tween, or PBS-T) and then blocked with 10% BSA in 0.1% PBS-T for 60 min at RT. Slides were incubated for 48 h at 4°C with rabbit anti-human $\alpha_v$ integrin (1:200) or both with rabbit anti-fd phage (1:500) and rat anti-mouse CD31 (1:50) antibodies diluted in 0.1% PBS-T with 1% BSA. Next, sections were washed and stained for 1 h at room temperature with the secondary antibodies AlexaFluor 488 goat anti-rabbit (1:500) and AlexaFluor 594 donkey anti-rat (1:500). Immunolabeling for the proliferation marker protein Ki67 was performed using a polyclonal primary rabbit antibody (1:400, Abcam). Analysis of apoptosis was done on frozen sections by immunostaining with an anti-caspase-3 rabbit antibody (1:1,000, Cell Signaling) that also detects the cleaved caspase-3. Immunostaining for caspase-3 was performed similar to Ki67 using a secondary biotinylated anti-rabbit antibody

(1:400). Both Ki67 and caspase-3 immunostainings were carried out on tumor serial sections, in similar tumor areas in each experiment, in order to test whether tumors are simultaneously undergoing reduction of proliferation and apoptosis. Staining for the stem cell marker CD133 in tumors was performed using an anti-CD133 monoclonal antibody (Miltenyi Biotec).

### Stem cell marker staining for FACS

We used primary monoclonal antibodies for Sox-2 (1:50, Biolegend), for CD133 (1:50, Miltenyi Biotec), and for Nestin (5 μg, R&D Systems).

### Statistical analysis

*P* values were generated by using either one-way ANOVA with Tukey's multiple comparison test (GraphPad Prism 6) or two-way ANOVA and Bonferroni's multiple comparison test, two-way ANOVA with Bonferroni correction (GraphPad Prism 6), and two-way ANOVA with Tukey's multiple comparison test (GraphPad Prism 6). In animal survival experiments, Kaplan–Meier method (Kaplan–Meier survival fractions) was used to generate *P* values and calculate the Log-rank (Mantel–Cox). Error bars represent standard error of the mean (SEM).

*In vitro* experiments were designed to ensure 5% significance level and a minimum of 80% power. Experiments were performed in triplicate and repeated, at least twice, increasing the total number of samples to $n = 6$, which is sufficient for a statistical power above 80%.

For animal experimentation, we used a randomized experimental design and blinding was also applied throughout the animal studies. Briefly, following tumor detection in mice by BLI of *Luc*, tumor-bearing mice were randomly assigned to experimental groups. Imaging of tumor-bearing mice was done, cage by cage, blindingly and by fixing the same parameters for all the cages. Histopathological analyses were also performed blindly. We have used 5–6 mice per group and repeated the experiments twice.

**Expanded View** for this article is available online.

### Acknowledgements

We thank Renata Pasqualini and Wadih Arap for the k91Kan *E. coli* bacteria and fUSE5 plasmid, Simona Parrinello, Nelofer Syed, Geoffrey Pilkington, and Steven Pollard for the human cells. This study was supported by a grant G0701159 of the UK Medical Research Council (MRC) and the Brain Tumour Research Campaign (BTRC). Keittisak Suwan is funded by the Children with Cancer UK (13/147 and 16/230). AMC is funded by ISCIII-FEDER (CP13/00189). JP received a stipend from the BTRC. EOA acknowledges support from Imperial College NIHR Biomedical Research Centre award (WSCC_P62585), Cancer Research UK grant (C2536/A16584), Medical Research Council grant (MC-A652-5PY80), and Experimental Cancer Medicine Centres grant (C37/A7283).

### Author contributions

JMP designed, optimized, analyzed the data, and performed all the *in vitro* experiments with the assistance of SW, MZIP, WY, GC, GS, AC, and KS. NGO and AMC provided the HSJD-GBM-001 cells and assisted in the design of *in vivo* experiments with this cell model. JMP, SW, KS, and AH performed all the *in vivo* experiments. GT edited the manuscript and supported JMP during the revision. EOA contributed to the study design, analysis, and interpretation of data. AH conceived, designed and funded the study, analyzed the data, and supervised all experiments.

### The paper explained

#### Problem

Glioblastomas are the most aggressive tumors of the central nervous system, and the least responsive to intervention, leading to death in a disproportionate number of patients under 40 years old. Thus, novel therapeutic approaches are urgently needed to effectively treat GBM or improve existing treatments. Gene therapy is promising in this disease, and GBM was the first to be treated by clinical gene therapy, but success has been limited by the inefficiency of vectors in delivering therapeutic genes at therapeutic levels within the tumors, and by the blood–brain barrier, requiring invasive intracranial delivery. This project aims to reprogram harmless bacteria viruses, named bacteriophage (phage), to deliver therapeutic nucleic acids to glioblastoma in a preclinical setting. Moreover, the phage was further modified to enhance therapeutic delivery in the presence of temozolomide (TMZ). These viruses can readily cross the physiological brain barriers and also target tumors specifically, leaving normal tissue unharmed.

#### Results

We now propose a novel bacteriophage-guided gene therapy vector to treat intracranial human GBM, in combination with the chemotherapeutic drug TMZ. We found that phage-guided gene therapy targets intracranial glioblastoma selectively following systemic administration through clinical non-invasive routes. Next, combination of phage-gene therapy with TMZ chemotherapy resulted in activation of phage-guided gene delivery and subsequently a synergistic action against human glioblastoma, leading to striking regression of these tumors in preclinical models.

#### Impact

Given the proven acceptability of temozolomide chemotherapeutic drug for human therapy against glioblastoma, as well as the accumulating evidence for the safety of bacteriophage in human and animals, these findings hold significant promise for the clinical efficacy of this combination therapy to treat the incurable glioblastoma.

### Conflict of interest

Amin Hajitou is inventor in a patent, US8470528 B2, relating to the AAVP technology. This patent was granted.

### For more information

https://www.invivogen.com/pdrive-grp78

http://www.biosci.missouri.edu/smithgp/PhageDisplayWebsite/PhageDisplayWebsiteindex.html

https://patents.google.com/patent/US8470528B2/en

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
