## [Review Process File · EMBO Molecular Medicine]

Efficacy of systemic temozolomide-activated phage-targeted gene therapy in human glioblastoma

Justyna Magdalena Przystal, Sajee Waramit, Md Zahidul Islam Pranjol, Wenqing Yan, Grace Chu, Aitthiphon Chongchai, Gargi Samarth, Nagore G. Olaciregui, Ghazaleh Tabatabai, Angel Montero Carcaboso, Eric Ofori Aboagye, Keittisak Suwan and Amin Hajitou

Review timeline:

Submission date:	15 September 2017
Editorial Decision:	20 September 2017
Additional correspondence:	9 February 2018
Additional correspondence:	6 April 2018
Revision received:	8 June 2018
Editorial Decision:	13 August 2018
Revision received:	20 December 2018
Editorial Decision:	22 January 2019
Revision received:	1 February 2019
Accepted:	4 February 2019

Editor: Céline Carret

Transaction Report:

1st Editorial Decision

20 September 2017

Thank you for the submission of your manuscript "Efficacy of systemically delivered temozolomide-activated phage-targeted gene therapy in human glioblastoma." We have now heard back from the three Reviewers whom we asked to evaluate your manuscript.

As you will see the reviewers find the study of interest, although they do express some concerns on your manuscript, which I would summarise as follows: 1) the conceptual advance of the study is not very high, while the synergistic effect reported is; one way to improve novelty would be to make the translational work even stronger by carefully assessing toxicity (referee 4) and answer to referee 2's clinical questions should the strategy move to the clinic; 2) improve conclusiveness by assessing overall survival of mice and testing primary human cells (referees 1 and 3).

After our reviewer cross-commenting exercise, there was full agreement that the above concerns (in addition to the other items) would need to be addressed to strengthen the data.

We would welcome the submission of a revised version within three months for further consideration and would like to encourage you to address all the criticisms raised as suggested to improve conclusiveness and clarity. Please note that EMBO Molecular Medicine strongly supports a single round of revision and that, as acceptance or rejection of the manuscript will depend on another round of review, your responses should be as complete as possible.

I look forward to receiving your revised manuscript.

***** Reviewer's comments *****

Referee #1

(Comments on Novelty/Model System for Author)

Since glioblastoma is sustained by a subpopulation of glioblastoma stem cells, endowed with stem cell properties and resistant to chemo and radiotherapy, Authors should assess the efficacy of their proposed treatment on patient-derived glioblastoma stem cells, both in vitro and in vivo, as a complementary approach to the use of glioblastoma cell lines.

(Remarks for Author)

This paper proposes a combinational therapy against Glioblastoma, using temozolomide together with a targeted suicide gene therapy that does not require intracranial delivery. The new vector described is a hybrid AAV/phage targeted to glioblastoma cells with a dual targeting system: i) the RGD4C ligand on its capsid binds specifically to $\alpha\beta$ integrin receptors, which are well expressed on glioblastoma stem cells and ii) the gene expression is under the control of Grp78 promoter, that is activated by the tumor microenvironment and boosted by Temozolomide treatment.

The feasibility of this novel approach has been well evaluated demonstrating that the vector is able to target intracranial tumors after intravenous administration, and there are preliminary promising results about its efficacy in vivo.

Still, the manuscript needs a revision:

1) In the introduction, Authors write that they used a RGD4C/AAVP-Grp vector expressing HSVtk; it should be better to explain the mechanism of action underlying the efficacy of the combination of a HSVtk-expressing vector and the administration of Ganciclovir.

2) According to a large amount of data present in literature, the HSVtk-dependent activation of Ganciclovir results in DNA synthesis impairment, thus leading to cell death. The nature itself of this mechanism of action makes it effective mainly against active-proliferating cells. Authors don't discuss the efficacy of this treatment against the GBM stem cell sub-population, a sub-set of slow-cycling cells endowed with stem cell like properties and able to survive to the standard treatment and to sustain the relapse. This is a big issue, since patient death is often consequent to GBM relapse. For this reason, it's strongly recommended to repeat in vitro and in vivo assays using primary GBM stem cells instead of cell lines, after assessing that GBM stem cells express $\alpha V\beta 3$ and $\alpha V\beta 5$ integrins.

3a) Although the in vivo data in terms of tumor volume reduction are promising, Authors should evaluate also the efficacy of the treatment on the overall survival of tumor-bearing mice.

3b) Moreover, to better understand the mechanisms underlying the in vivo efficacy of the treatment, Authors should evaluate and quantify the presence of proliferation and apoptosis markers in the brain tumors, by mean of IHC analysis.

4) According to Fig.2B, the Luciferase production upon TMZ treatment is delayed in SNB19 compared to the other cell lines, suggesting a slower activation of Grp78 promoter. This is coherent with the Fig. 3, showing a weaker and slower TMZ-dependent activation of the UPR pathway (that regulates Grp78 activation) in SNB19 compared to U87 and LN229.

Surprisingly, as shown in fig.4, SNB19 cells are the most responsive, in vitro, to the addition of TMZ, since only a modest efficacy of the combination of TMZ+GCV compared to GCV alone is shown for the other cell lines.

Authors should try to discuss this apparent contradiction.

5) Authors should provide new images of the Western Blot shown in the Fig.2, because, especially for SNB19, GAPDH and Grp78 bands seem to have run in opposite directions.

6) For clarity's sake:

- vectors should be always indicated in the same way (sometimes they are indicated with HSVtk, sometimes HSVtk is omitted, for instance).
- a figure describing the scheme of the used vector is recommended
- in the Fig.6 legend and in the text, Authors explain that mice have been treated with GCV and/or TMZ. However, the treatment with GCV is not indicated in the Fig. 6.

Referee #3

(Comments on Novelty/Model System for Author)

(Remarks for Author)

1. The strategy of using the capsid of M13 phage to deliver a recombinant AAV genome is an interesting approach to treating glioblastoma, with the phage capsid expressing the RGD4C ligand that binds the avb3 integrin receptor, and an inducible promoter to deliver a suicide gene. The biggest challenge of moving this strategy to the clinic is delivery to the tumor cells and efficiency of inducing the promoter.
2. Re the overall strategy, TMZ is a chemotherapeutic agent. What if it cannot be used (for example toxicity) at all times when the physician would like to turn on the phage delivered genome?
3. The strategy is use the recombinant phage intravenously. What percentage of the delivered dose reaches the CNS? What percentage reaches the GBM?
4. The studies were carried out in immunodeficient rodents; what is the immunogenicity of the recombinant phage in wild type rodents?
5. The 3 tumor lines used were screened for the avb3 integrin; what % of glioblastoma express this integrin to the same level? This is important regarding how useful the strategy will be in real life.
6. How does the levels of TMZ scale to possible use in humans?
7. The phage was observed in brain blood vessels and tumors; where else?
8. Was the therapy absolutely specific for GBM? Was there any adverse effects in normal brain or systemic organs?
9. Were any studies carried out with primary GBM?
10. What was the mortality of the animals vs controls?

Referee #4

(Comments on Novelty/Model System for Author)

- a) The vast majority of the findings have been already described in previous publications.
- b) The Authors need to address the toxicity of the proposed treatment on a large panel of human primary cells and cell lines.

(Remarks for Author)

In the manuscript from Przystal et al., entitled "Efficacy of systemically delivered temozolomide-activated phage-targeted gene therapy in human glioblastoma", the Authors set-up a strategy to treat human glioblastoma multiforme (GBM) in an orthotopic xenograft model by combining hybrid AAV/M13-phage vector, as a tool to deliver and express toxic genes specifically to GBM cells, and temozolomide (TMZ) administration.

The system is based on a recombinant adeno-associated virus genome (rAAV) inserted within the M13 bacteriophage genome that can be thus incorporated in the M13 capsid. This vector, when injected systemically is able to pass the blood-brain barrier and transduce brain cells. The M13 capsid used was engineered to display the CDCRGDCFC (RGD4C) ligand that binds specifically the heterodimer $\alpha\beta3$ integrin cell surface receptor, which is overexpressed on human GBM. To further increase the specificity of expression of toxic genes in GBM cells the Authors adopted a tumor-activated and temozolomide (TMZ)-induced promoter of the glucose-regulated protein 78 (Grp78) gene. The Authors TMZ increases endogenous Grp78 gene expression and boosts transgene expression from the new vector construct (RGD4C/AAVP-Grp78) and targets intracranial tumors in mice following intravenous administration. Finally the Authors show that TMZ amplified tumor cell killing in combination with RGD4C/AAVP-Grp78-HSVtk expressing the Herpes simplex virus type-I thymidine kinase (HSVtk) when combined with GCV *in vitro*. The Authors conclude that this combined treatment display a synergistic effect to suppress growth of orthotopic glioblastoma.

Remarks

The manuscript is well written and the claims are sustained by convincing data. The novelty of this manuscript is in the finding that the TMZ and RGD4C/AAVP-Grp78-HSVtk combined treatment display a synergistic effect to suppress growth of orthotopic glioblastoma, while the other findings have been already reported with more or less details in other journals.

The main limitation of manuscript is that no data regarding the safety and specificity of the treatment is provided. Although the targeting appears to be very specific for the 3 different human GBM cell lines analyzed, while mouse cells are not transduced, the study does not address the impact of the treatment on other human cell types. Indeed, the RGD4C ligand was selected to bind specifically the heterodimer $\alpha\beta3$ integrin in human cells which is expressed in a wide variety of cells and not only GBM. What is the effect of the treatment on other human cell types? This is an outstanding issue since this treatment could result in an extreme toxicity (potentially lethal) to the patients. The Authors should address this outstanding issue experimentally on a large panel of human primary cells and cell lines.

Additional author correspondence

9 February 2018

Many thanks for providing us with the opportunity to submit a revised manuscript. We are pleased that the reviewers found our study interesting. Their comments and advice have been helpful to improve the manuscript. I am writing to request your approval to provide us with an extension for the submission of our revised manuscript, beyond the initial three months.

Brief, to date we have performed extensive work and generated some exciting and novel data. To comply with the reviewers, we have made the necessary efforts and established new collaborations in order to gather numerous primary GBM as well as GBM-derived stem cells to initiate the work, as we didn't have these cells in house or in our institution. We managed to receive all the cells in December 2017 after all the MTAs were processed. Then we had to screen the new cells for expression of the $\alpha\text{-v}$ integrin receptors of RGD4C/phage-Grp78 vector. Next, we carried out extensive work to confirm transduction of these cells by the RGD4C/phage-Grp78 and investigated the temozolomide (TMZ) activation of this vector in a list of primary and GBM-derived stem cells, by using increasing doses of TMZ. Importantly, primary human normal cells showed very low levels of expression of integrin receptors and insignificant transduction by the RGD4C/phage-Grp78. We are currently combining RGD4C/phage-Grp78 with increasing concentrations of TMZ to rule out any TMZ-activation of the vector in these normal cells.

Now we would like to wrap up our *in vivo* studies. Actually, before initiation of *in vivo* experiments, we labelled primary tumour cells with a luciferase expressing lentiviral vector so that we can image intracranial GBM with bioluminescent imaging of luciferase. It is well known that gene transfer to primary tumour cultures can be difficult and require time to optimise. Then, we screened primary cells *in vivo* to identify cells that can establish intracranial tumours. The growth of these cells *in vitro*, in particular the lentiviral-infected cells, is slower than that of commercial tumour cell lines, which takes us a lot longer to have enough cells to implant into mice. It could take up to a month from lentiviral vector infection of cells to growing the amount of cells for an *in vivo*

experiment. Altogether, these experiments have also required massive viral vector production, purification and sequencing to confirm the intact display of the RGD4C ligand on the pIII minor coat protein.

We would be grateful for an extension to accomplish the *in vivo* work, our comprehensive *in vitro* investigation points out to a promising *in vivo* outcome. Additionally, most of my team were away during Christmas College closure.

Additional author correspondence

6 April 2018

I am writing in regards to the submission of our revised manuscript. We have successfully completed all *in vitro* and *ex vivo* studies. We have also performed *in vivo* experiments. Now we are analysing the *in vivo* data and also repeating some of the *in vivo* experiments in order to confirm our findings in tumor-bearing mice.

Therefore, I am writing to request an additional two-month extension for the submission.

1st Revision - authors' response

8 June 2018

We are delighted to resubmit a revised manuscript to address the reviewers' comments and further the findings of the study. In addition to revision of the text, we performed multiple additional experiments and included key information. We have addressed the reviewers' comments point-by-point in the text below. We have also highlighted the changes, in yellow, in the revised manuscript to make them clearly visible to the reviewers.

Reviewer #1

We thank reviewer#1 for a comprehensive review of our manuscript. We appreciate the comment: "The feasibility of this novel approach has been well evaluated demonstrating that the vector is able to target intracranial tumors after intravenous administration, and there are preliminary promising results about its efficacy *in vivo*".

We have addressed her/his specific comments as follows:

1. In the introduction, Authors write that they used a RGD4C/AAVP-Grp vector expressing HSVtk; it should be better to explain the mechanism of action underlying the efficacy of the combination of a HSVtk-expressing vector and the administration of Ganciclovir.

Response:

We used the *HSVtk* mutant SR39 that has increased sensitivity to ganciclovir (GCV) when compared to wild-type *HSVtk* (Black et al, 2001). To comply with the reviewer, we have added an explanation of the mechanism of action of the *HSVtk*-expressing vector plus ganciclovir, and stated the use of the *HSVtk* mutant S39, in the introduction on pp. 4-5.

Briefly, the *HSVtk* enzyme phosphorylates prodrug nucleoside analogues such as GCV, and converts them into nucleoside analogue monophosphates, which are subsequently converted into triphosphates. These triphosphate GCV compounds are then incorporated into the cellular genome, inhibit DNA polymerase and subsequently induce cell death by apoptosis (Hamel et al, 1996). It is important to note that the *HSVtk*/GCV approach also elicits a bystander effect, which means that cells containing the *HSVtk* kill neighbouring non-transduced cells through the gap junctional intercellular communications (GJIC) that allow the transfer of the converted cytotoxic drug and/or toxic metabolites between these cells as we have previously reported (Trepel et al, 2009). After a few days, this bystander effect results in increased cell killing by the RGD4C/AAVP-*HSVtk* vector. This "bystander effect" may potentially overcome the requirement for all malignant cells to be transduced in order to achieve meaningful tumor regression (Culver et al, Science 1992, Grignet-Debrus et al, 2000).

2. According to a large amount of data present in literature, the HSVtk-dependent activation of Ganciclovir results in DNA synthesis impairment, thus leading to cell death. The nature itself of this mechanism of action makes it effective mainly against active-proliferating cells. Authors don't discuss the efficacy of this treatment against the GBM stem cell sub-population, a sub-set of slow-cycling cells endowed with stem cell like properties and able to survive to the standard treatment and to sustain the relapse. This is a big issue, since patient death is often consequent to GBM relapse. For this reason, it's strongly recommended to repeat *in vitro* and *in vivo* assays using primary GBM stem cells instead of cell lines, after assessing that GBM stem cells express $\alpha_v\beta_3$ and $\alpha_v\beta_5$ integrins.

Response:

We thank the reviewer for raising this important point. To address the reviewer's suggestions, we performed several experiments on GBM stem cells (GSC) that we received from the collection of Dr Steven Pollard, expert in Neural Stem cells and Brain Cancer, from the Edinburgh Brain Cancer Center. We have performed experiments using two primary GSCs named G26 and G166 (Pollard 2013).

First, immunofluorescent staining using antibodies against the integrin subunits α_v , β_3 and β_5 that form the two heterodimers $\alpha_v\beta_3$ and $\alpha_v\beta_5$ showed expression of these three integrins by both G26 and G166 primary stem cells.

Second, treatment of G26 and G166 stem cells with RGD4C/AAVP-*Luc* carrying the luciferase (*Luc*) reporter gene showed efficient gene delivery to GBM stem cells. Moreover, no gene delivery was obtained with the control non-targeted/AAVP-*Luc* vector lacking the RGD4C ligand, proving that delivery to human GBM primary stem cells by RGD4C/AAVP is selective and mediated through binding of RGD4C ligand to the integrin receptors.

Third, addition of temozolomide (TMZ) enhanced gene delivery by RGD4C/AAVP-*Grp78-Luc* in both GSC in a dose dependent manner without affecting the vector selectivity. These data prove the ability of TMZ to activate the RGD4C/AAVP-*Grp78* vector in GBM stem cells.

Finally, we evaluated cell killing of GSCs and found that these cells are resistant to TMZ, which is part of standard treatment, consistent with the reviewer's point above. Importantly, treatment with the RGD4C/AAVP-*HSVtk* vector carrying the *HSVtk* gene induced substantial destruction of these stem cells in the presence of GCV, which was further and significantly pronounced in the presence of TMZ. These data are now included in the Results Section of the revised manuscript, on pp. 12-13 and Fig 7.

We did not complete our *in vivo* therapy on primary GSCs as these cells were not consistent in inducing tumors in immunodeficient mice, as well as in tumor initiation since mice started tumors at various times following cell implantation and after a longtime. Therefore, regrettably, a comprehensive *in vivo* evaluation using GBM stem cells will require more time and resources for future studies. Nonetheless, the present *in vitro* findings clearly show proof-of-efficacy of gene delivery to primary GSCs and their destruction by the vector, then establish a comprehensive *in vitro* foundation for the application of RGD4C/AAVP-*Grp78* vector as a platform for gene delivery to GBM stem cells.

To further strengthen our findings on primary cells, and as requested by another reviewer, we carried out *in vivo* evaluation on primary GBM tumor cells implanted intracranial in immunodeficient mice. First, we repeated the experiments above on human primary GBM cells from different sources; SEBTA-023 cells from The Portsmouth University and HSJD-GBM-001 from the collection of Dr Carcaboso, Hospital Sant Joan de Deu Barcelona, Spain. These primary GBM cells express the integrin subunits α_v , β_3 and β_5 . The vector efficiently delivered the *Luc* reporter gene to the cells, in a targeted manner.

Next, *in vivo* experiments were carried out using primary HSJD-GBM-001, and showed that intravenous administration of RGD4C/AAVP-*Grp78-HSVtk* in combination with GCV resulted in i) tumor growth inhibition, ii) increased survival of GBM-bearing mice, iii) decreased Ki67 cell proliferation marker, and iv) induction of apoptosis in primary GBM. Importantly these effects were

further enhanced when the vector was combined with TMZ. These data are shown in the results section of the revised manuscript on page 11, pp. 13-14 and Figs 6 & 8.

3a. Although the in vivo data in terms of tumor volume reduction are promising, Authors should evaluate also the efficacy of the treatment on the overall survival of tumor-bearing mice.

Response:

We have performed new experiments to evaluate efficacy of treatment on survival of mice with either intracranial U87 tumors or intracranial primary GBM (HSJD-GBM-001). All the treatments-RGD4C/AAVP-Grp78-HSVtk vector plus GCV, TMZ alone, or combination of RGD4C/AAVP-Grp78-HSVtk/GCV with TMZ improved the survival of tumor-bearing mice for both U87 and primary GBM. Treatment with RGD4C/AAVP-Grp78-HSVtk vector plus GCV generated better survival benefit for both U87- and primary GBM-bearing mice when compared to treatment with the non-targeted/AAVP-Grp78-HSVtk vector plus GCV. Importantly, tumor-bearing mice that received combination treatment of RGD4C/AAVP-Grp78-HSVtk/GCV with TMZ showed the highest survival benefit as compared to TMZ or RGD4C/AAVP-Grp78-HSVtk vector plus GCV. These results are now included in the revised manuscript on pages 10 and 13, and in Figs 4 & 8.

3b. Moreover, to better understand the mechanisms underlying the in vivo efficacy of the treatment, Authors should evaluate and quantify the presence of proliferation and apoptosis markers in the brain tumors, by mean of IHC analysis.

Response:

We have complied with the reviewer and evaluated the effects of treatments on apoptosis and proliferation in both intracranial U87 and primary HSJD-GBM-001. While apoptosis was performed on U87 tumors recovered at the end of the experiment, this was done on the HSJD-GBM-001 tumors recovered before the end of the experiment, at day 12 post treatment initiation, to avoid the apoptosis seen in the large U87 control tumors. We used an anti-caspase-3 antibody that also detects the cleaved caspase-3 and marks apoptotic cells because the HSVtk/GCV strategy is associated with apoptotic death of cells. We evaluated the effect on cell proliferation by staining for the proliferation marker protein Ki67. The combination treatment of TMZ with RGD4C/AAVP-Grp78-HSVtk/GCV induced the highest level of apoptosis in both types of glioblastoma and generated the greatest reduction in cell proliferation compared to monotherapies of TMZ alone or RGD4C/AAVP-Grp78-HSVtk/GCV. These data are now included in the results section of the revised manuscript on pp. 10-11, pp. 13-14 and Figs 5 & 8.

4. According to Fig.2B, the Luciferase production upon TMZ treatment is delayed in SNB19 compared to the other cell lines, suggesting a slower activation of Grp78 promoter. This is coherent with the Fig. 3, showing a weaker and slower TMZ-dependent activation of the UPR pathway (that regulates Grp78 activation) in SNB19 compared to U87 and LN229.

Surprisingly, as shown in fig.4, SNB19 cells are the most responsive, in vitro, to the addition of TMZ, since only a modest efficacy of the combination of TMZ+GCV compared to GCV alone is shown for the other cell lines.

Authors should try to discuss this apparent contradiction.

Response:

First, the reviewer is totally right about the delayed activation of the Grp78 promoter by TMZ in SNB19 cells. In attempt to understand this late response to TMZ, we compared the endogenous promoter activity between the three LN229, U87 and SNB19 cell lines by evaluating expression of the endogenous Grp78 protein by Western blot. Interestingly, the SNB19 cells had the highest expression of Grp78. This could mean that it is possible that the SNB19 cells initially had higher basal promoter activity of Grp78, and hence these cells did not need an early activation of Grp78, but required more time to achieve induction by TMZ.

As per the highest response of SNB19 cells *in vitro* to the addition of TMZ, first it is important to mention that in the absence of TMZ, these cells also show the most response of Grp78 promoter to addition of GCV. Therefore, it is possible that the highest response of SNB19 cells to TMZ plus GCV combination might, at least in part, be associated with the elevated response of Grp78

promoter, in these cells, to GCV treatment. Then the question arose as to why these cells are more sensitive to RGD4C/AAVP-Grp78-HSVtk/GCV. This could be due to GCV-mediated activation of the Grp78 promoter as we previously reported that HSVtk/GCV therapy upregulates Grp78 and transgene expression via the conserved unfolded protein response (UPR) signalling cascade (Kia et al. 2012).

On the other hand, cell death was measured at day 4 post GCV and TMZ treatments, which is 3 days after GRP78 induction by TMZ. Therefore, it is also possible that, at day 4, the bystander effect prompted by the HSVtk/GCV could take over, resulting in increased cell killing by the RGD4C/AAVP-HSVtk plus GCV. Thus, the cell killing data should be partially proportional to the levels of induction of Grp78 promoter. To confirm the bystander effect in the cells, we performed a Western blot for the main connexins involved in the bystander effect, in LN229, U87 and SNB19 cells, and found that all these cell lines express connexin-26. Connexins are proteins composing the channels of the GJIC through which toxic phosphorylated ganciclovir and/or other toxic intracellular metabolites are exchanged between one cell and another.

To recap, SNB19 cells could have higher sensitivity for Grp78 activation by HSVtk/GCV treatment, which was further enhanced by combination of GCV with TMZ.

To clarify this, we have added a discussion in the revised manuscript on page 16 and also included the connexin data as supplementary information in Fig EV4.

5. Authors should provide new images of the Western Blot shown in the Fig.2, because, especially for SNB19, GAPDH and Grp78 bands seem to have run in opposite directions.

Response:

We apologize to the reviewer for this. We have provided new images of the SNB19 Western Blots in the new Fig 1C of the revised manuscript.

5) For clarity's sake:

- vectors should be always indicated in the same way (sometimes they are indicated with HSVtk, sometimes HSVtk is omitted, for instance).

Response:

We have complied with the reviewer and indicated the vectors in a consistent manner.

- a figure describing the scheme of the used vector is recommended

Response:

We have complied with the reviewer and added a description of the vector's scheme in Fig EV1.

- in the Fig.6 legend and in the text, authors explain that mice have been treated with GCV and/or TMZ. However, the treatment with GCV is not indicated in the Fig. 6.

Response:

We apologise that GCV was not also added into the Fig.6; we have now done this to make the legends clear in the new Figs 4-5, Fig 8 and Fig EV2 of the revised manuscript.

Reviewer #3

1. The strategy of using the capsid of M13 phage to deliver a recombinant AAV genome is an interesting approach to treating glioblastoma, with the phage capsid expressing the RGD4C ligand that binds the avb3 integrin receptor, and an inducible promoter to deliver a suicide gene. The biggest challenge of moving this strategy to the clinic is delivery to the tumor cells and efficiency of inducing the promoter.

Response:

We would like to thank the reviewer for a critical review of our manuscript, we appreciate the comment "The strategy of using the capsid of M13 phage to deliver a recombinant AAV genome is an interesting approach to treating glioblastoma..."

Delivery to the tumor cells and efficiency of inducing the *Grp78* promoter are addressed with the next reviewer's points.

2. Re the overall strategy, TMZ is a chemotherapeutic agent. What if it cannot be used (for example toxicity) at all times when the physician would like to turn on the phage delivered genome?

Response:

We totally agree with the reviewer that TMZ cannot be given to patients at all times. Indeed, TMZ is discontinued in patients with Absolute Neutrophil Count $< 500/\text{mm}^3$, platelets $< 10,000/\text{mm}^3$ and Common Toxicity Criteria Grade 3 or 4 non-hematologic toxicity. Luckily the *Grp78* promoter can be induced by various other therapeutic approaches e.g radiation therapy (Sun et al, 2017) and other chemotherapeutic drugs used for brain tumor treatment such as cisplatin (Mandic et al, 2003). Moreover, we recently found that a safe anti-cancer agent, curcumin from natural sources, has the potential to induce the *Grp78* promoter from the RGD4C/AAVP-*Grp78* vector, in human primary gliomas, consistent with previous studies reporting the ability of curcumin to induce the *Grp78* gene in cancer cells through the UPR pathway (Kim et al, 2016).

Curcumin, derived from the *Curcuma longa* plant, constitutes a commonly used Southeast Asian spice, turmeric. It is widely used across the region as a spice and for its medicinal anti-inflammatory properties as an ointment (Park et al, 2013). New studies have highlighted curcumin's potential as an anti-cancer agent, due to its ability to induce tumour cell death with no systemic side effects. Furthermore, *in vitro* studies have shown that curcumin is able to prevent proliferation of cancer cell lines including glioblastoma multiforme (Klinger et al, 2016). Additionally, curcumin mediates pathways which promote apoptosis in cancer cells. Importantly, curcumin has been shown to cross the blood-brain barrier (Klinger et al, 2016); and *in vitro* studies using a glioblastoma cell line found curcumin treatment resulted in decreased cell proliferation, DNA fragmentation and apoptosis through a pathway involving caspase-3, 8 and 9 (Klinger et al, 2016; Huang et al, 2010).

In this revised manuscript, we have not included our data with the curcumin effect on the RGD4C/AAVP-*Grp78* but enclosed them below (Figure-1), in this rebuttal letter, for the reviewer's information. However, we have added a brief discussion about this in the revised version of this manuscript on page 15.

Figure 1- Paediatric human primary DIPG cells (diffuse intrinsic pontine glioma) were seeded in 96-well plates until 70-80% confluent, then transduced with 1×10^6 TU of targeted RGD4C/AAVP-*Grp78*-Luciferase or non-targeted/AAVP-*Grp78*-Luciferase control vector. At day 3 post-transduction, concentrations of curcumin ($0 \mu\text{M}$ - $40 \mu\text{M}$) or carboplatin ($0 \mu\text{M}$ - $100 \mu\text{M}$) were added. At day 6 post-transduction, Luminescence was measured and normalised to untreated and non-

*transduced control cells. Results are shown as mean \pm SEM of triplicate wells. Luminescence (RLU) is a measure of Luciferase expression in cells. Triplicate repeats of the experiment were conducted. * $p < 0.05$ and ** $p < 0.01$ are marked as statistically significant.*

3. The strategy is use the recombinant phage intravenously. What percentage of the delivered dose reaches the cns? What percentage reaches the gbm?

Response:

The ability of phage to cross the BBB and delivery to the brain was described as early as 1943 by Dubos et al. (Dubos et al, 1943) in an investigation to find a cure against meningitis. First, they performed pharmacokinetic studies in uninfected mice and showed immediate phage apparition in the bloodstream whereby levels dropped sharply within hours, and very few were ever seen in the brain. Importantly, in infected animals with a bacterial infection in the brains, brain levels of phage largely exceeded the blood levels (10^7 - 10^9 phage/g of brain tissue) by 8 hour which started to decrease only between 75 to 138 hr, following infection elimination. These findings provide evidence that phages cross the BBB, and that phage gets rapidly cleared from the brain in the absence of its bacterial host, likely by microglial cells as phage is well known to get cleared by the reticulo-endothelial system.

Then, following introduction of the *in vivo* phage display technology, the M13 Phage (parent of the AAV/phage) was used to identify targets in the brain by *in vivo* screening, which comprised of the intravenous delivery of phage display peptide libraries of up to 1×10^9 different peptides. These *in vivo* screening studies allowed isolation of phage peptide clones that bind to target receptors within the brain (Li et al, 2012).

In conclusion, it appears that the phage gets rapidly cleared from the brain unless a target is present, such as host bacteria or a mammalian receptor (i.e. integrin), which is consistent with the ability of *in vivo* screening with phage display ligand libraries to select for brain homing phage clones that bind to receptors in the brain. This explains why we detect insignificant levels phage in the healthy brains as they do not express the integrin receptors of the RGD4C/AAVP-*Grp78*, even at 18 hours post vector delivery at which time points the brain levels should be at their peak based on Dubos study 1943. Whereas, we always recovered high levels from brain tumors that express the integrin receptors. This is not specific to RGD4C, as our findings are consistent with other groups using other ligands. For instance, a previous elegant study clearly showed the phage ability to cross the BBB in intracranial human gliomas in mice following IV administration (Ho et al, 2004; Ho et al, 2010). The authors used a glioma-binding bacteriophage based on the M13 phage, parent of AAVP, that was injected intravenously (IV) into mice with intracranial human glioma and allowed to circulate for 24 hours. Phage titration in tissues revealed specific phage homing to intracranial human gliomas upon IV delivery, but no detectable levels of phage were recovered from the healthy brains.

To answer the reviewer, we provide detailed data of phage levels recovered from intracranial GBM and healthy brains following IV delivery, as follow:

- From an average of 0.13 g of tumor tissue, we recover an average of $\sim 0.2 \times 10^6$ functional and intact RGD4C/AAVP TU by bacterial infection, following intravenous administration of 5×10^{10} functional TU per mouse. This does not take into account the non-recovered phage, non-infectious and unable to infect bacteria, that has initiated processing by the cells, and which could still deliver the therapeutic AAV genome to the cells.
- In the same experiments from an average of 0.13 g of the healthy brain tissue, we recover an average of $\sim 19,000$ functional and intact RGD4C/AAVP TU, following intravenous administration of 5×10^{10} TU per mouse.

4. The studies were carried out in immunodeficient rodents; what is the immunogenicity of the recombinant phage in wild type rodents?

Response:

Phages are not completely ignored by the mammalian immune system, which will eventually sequester and clear them by the reticulo-endothelial system (Geier et al, Nature 1973). The M13 phage, parent of AAVP, was well reported for its immunogenicity in wild type rodents (Trepel et al, 2001).

We also showed that AAVP viral particles are immunogenic in our therapy studies in tumor-bearing mice vaccinated against the RGD4C/AAVP virus (Hajitou et al, 2006). Then, an independent study under the direction of the National Cancer Institute NCI USA, carried out by the Vets, confirmed the immunogenicity of RGD4C/AAVP-TNF α , carrying the cytokine Tumor Necrosis Factor Alpha (TNF α) in wild type pet dogs with natural tumors (Paoloni et al, 2009). In humans, the immunogenicity was confirmed by using the M13 phage (parent of the RGD4C/AAVP), in the clinical trial published in (Krag et al, 2006) and in which a phage library, based on M13 phage, was administered intravenously to cancer patients that generated an immune response against phage. It is noteworthy to mention some attractive features of the phage, yielded by the studies cited above, and that are related to the safety of repeated dosing of phage and its efficacy against cancer. Briefly, repeated administrations of RGD4C/AAVP-*HSVtk* plus GCV showed anti-tumor efficacy in immunocompetent mice despite the presence of IgGs against the phage capsid (Hajitou et al. Cell 2006). Moreover, repeated injections of RGD4C/AAVP-*HSVtk* plus GCV remained surprisingly effective against tumors in phage-vaccinated immunocompetent mice despite very high titers of circulating anti-phage IgG (Hajitou et al, Cell 2006). Next, in the pet dog study performed under the direction of the NCI USA, we reported safety of single and repeated dosing of the RGD4C/AAVP-TNF α in wild type pet dogs with natural tumors (Paoloni et al, 2009), with no maximally tolerated dose reached despite the presence of circulating IgGs against the phage. Remarkably, repeated weekly intravenous injections resulted in eradication of aggressive tumors in a few pet dogs (Paoloni et al, 2009). In the clinical trial by (Krag et al, 2006), no serious clinical side effects, including allergic reactions, were observed with serial up to three intravenous infusions, despite the presence of anti-phage IgGs. Importantly, phages were successfully recovered from tumors of every patient that underwent infusion, and phage recovery was increased with increasing doses.

To add to this, in our initial studies published in Cell 2006, we showed that when tumors grew back after termination of therapy, repeated administrations of RGD4C/AAVP-*HSVtk* again inhibited tumor growth in wild type mice and improved survival of tumor-bearing mice. Phage-based particles are known to be immunogenic, but this feature can be modulated through targeting itself (Trepel et al, 2001).

5. The 3 tumor lines used were screened for the $\alpha v \beta 3$ integrin; what % of glioblastoma express this integrin to the same level? This is important regarding how useful the strategy will be in real life.

Response:

The $\alpha v \beta 3$ integrin is highly expressed in GBM. In view of the reviewer's comment we have added the following statements:

“Regarding the relevance of this work, 55% or 12% of glioblastoma tissues express $\alpha v \beta 3$ when scored as ‘mild’ or ‘moderate to strong’, and expression is seen on both endothelial and glial cells; in contrast, $\alpha v \beta 3$ expression is restricted in low grade astroglial-derived tumors, reactive astrogliosis, or on glia/neurons in normal adult cortex and cerebral white matter, and increases in relation to malignancy in gliomas (Gladson and Cheresch 2001; Schnell et al. 2008).

6. How does the levels of TMZ scale to possible use in humans?

Response: We have added the following statement to clarify this point:

“We use TMZ administered daily by i.p. injection at 30mg/kg ($\sim 90\text{mg/m}^2$) in mice. This was intended to achieve approximately similar dose levels as humans (typically 75 mg/m² p.o. or i.v. daily for 42 days concomitant with focal radiotherapy followed by a higher maintenance dose of 150 - 200 mg/m²) “

7. The phage was observed in brain blood vessels and tumors; where else?

Response:

We previously investigated the localisation of the RGD4C/AAVP vector within the tumors following intravenous administration to tumor-bearing mice (Hajitou et al, 2006; Tandle et al, 2009) or to pet dogs with spontaneous natural tumors (Paoloni et al, 2009). In those studies, we either used antibodies against the phage capsid in immunostaining experiments to detect the localisation of the phage particles within the tumors or examined the localisation of AAVP-mediated gene expression, by using vectors carrying the GFP reporter. While the majority of the RGD4C/AAVP particles were localised within the tumor vasculature, we also detected vector particles in cells surrounding the

blood vessels. This could be explained by the fact that the tumor blood vessels are readily accessible to the vector through the systemic circulation, and this is consistent with the fact the *in vivo* phage display technology with M13 phage display peptide libraries in tumor-bearing mice has mostly identified vascular receptors in these preclinical models. There is also the tumor interstitial pressure and extracellular matrix that could hinder phage entry into the tumor stroma (Yata et al, 2015). On the other hand, the localisation of RGD4C/AAVP vector outside the tumor blood vessels could be facilitated by the leaky tumor vasculature allowing the phage to diffuse to transduce cells expressing the integrin receptors; or because phage could also target these cells by its ability to cross vessel barriers. We believe that these cells should be tumor cells, and this was strengthened by the ability of RGD4C/AAVP-GFP vector to express GFP in cells surrounding the blood vessels (Hajitou et al, 2006). We also believe that phage can also be internalised by the immune system infiltrating the tumors. Indeed, a small percentage of activated leukocytes and macrophages can express low levels of $\alpha_v\beta_3$ integrin, and phage is known to be taken up the Antigen Presenting Cells (APC) which will process the phage, then the tumor exposure of phage will attract these APC that would infiltrate the tumors. However, our *in vitro* studies showed that the internalised phage was unable to generate gene expression in these immune cells.

We agree with the reviewer that it is important to uncover the exact cell types that internalise the RGD4C/AAVP vector in addition to the tumor vasculature. However, we believe this is a separate and important project to investigate in the future, as it could enhance our understanding of the therapeutic mechanism of this vector and further improve its efficacy.

8. Was the therapy absolutely specific for gbm? Was there any adverse effects in normal brain or systemic organs?

Response:

First, there is a large body of literature reporting the safety of RGD4C/AAVP in rodents and large animals, and of the M13 phage, parent of RGD4C/AAVP, in humans. Moreover, to date in all the pre-clinical models tested, biodistribution of gene delivery by the RGD4C/AAVP vector generated gene expression in tumors exclusively without any detectable gene expression in the healthy tissues. In tumor-bearing mice, in addition to non-targeted AAVP, additional controls such as scrambled AAVP or mutated RGE/AAVP did not generate any gene delivery to tumors proving the tumor selectivity of the RGD4C ligand (Hajitou et al, 2006).

In tumor-bearing mice, as we previously reported (Hajitou et al, 2006), biodistribution using bioluminescent imaging of luciferase in living tumor-bearing mice with vectors carrying the *Luc* reporter gene, showed specific expression of the *Luc* reporter gene in tumours with no expression detected in the healthy tissues, these data were confirmed with GFP vectors. In the same work (Hajitou et al. 2006), we reported an elegant study using micro-PET imaging of *HSVtk* with the radiolabelled nucleoside analog [^{18}F]FEAU (2'-[^{18}F]-fluoro-2'-deoxy-1- β -D-arabino-furanosyl-5-ethyl-uracil), radiolabelled substrate of the *HSVtk* enzyme and showed specific accumulation of [^{18}F]FEAU in the tumors of mice, and subsequently selective expression of the *HSVtk*, with no accumulation of the [^{18}F]FEAU in the healthy tissues. Histopathological analyses of the systemic organs and brains removed from tumor-bearing mice treated by the same experimental protocol revealed no histopathologic abnormalities. Then in 2009, an independent group (Tandle et al. Cancer 2009) used the RGD4C/AAVP for targeted gene therapy with the cytokine, $\text{TNF}\alpha$ gene, and they reported specific expression of $\text{TNF}\alpha$ in the tumors following intravenous administration to tumor-bearing mice, with no $\text{TNF}\alpha$ detected in the healthy tissues.

In larger animals, a study carried out under the direction of the National Cancer Institute of the USA elegantly demonstrates the safety and potential of this technology. Targeted RGD4C/AAVP was used to deliver $\text{TNF}\alpha$ to naturally occurring cancers diagnosed in dogs (Paoloni et al. PlosOne 2009). In this study, intravenous infusion of the targeted RGD4C/AAVP- $\text{TNF}\alpha$ generated selective accumulation of the vector particles in the tumors, with no vector detected in the healthy tissues, and this tumor accumulation of the vector translated into selective expression of $\text{TNF}\alpha$ in the tumors, exclusively. Importantly, dose escalation and repeated RGD4C/AAVP vector dosing proved safe. Intravenous administration of RGD4C/AAVP- $\text{TNF}\alpha$ dose up to 10^{13} TU/dog was safe, and this dose is equivalent to the 5×10^{10} TU/mouse dose that we tested in our current study on mice with GBM. Since no maximal tolerated dose (MTD) was achieved, in dogs, it is possible that higher doses of RGD4C/AAVP- $\text{TNF}\alpha$ may also be safely administered.

In the clinical study published by (Krag et al. Cancer Res. 2006), M13 phage display libraries were injected to eight patients with stage IV cancer, including melanoma, breast and pancreatic cancers. The phage library, based on M13 phage (parent of the RGD4C/PAAV), was administered intravenously in order to identify tumor-targeted M13 phages. No serious clinical side effects, including allergic reactions, were observed with serial up to three intravenous M13 phage infusions, despite the presence of anti-phage IgGs. In that trial, patients received 1.6×10^8 to 1.0×10^{11} TU/kg, equivalent of an approximate range of 1.0×10^{10} to 1×10^{13} TU per patient. These phage doses were able to successfully target the tumors following intravenous administration.

In addition to the literature cited above, we further complied with the reviewer and performed additional experiments. First, we measured the animal weight during the course of therapy, in mice bearing either intracranial U87 or primary intracranial GBM. We found that animals did not show any weight loss related to treatment with the combination of RGD4C/AAVP and TMZ and any weight loss at the end of experiments was the consequence of the intracranial tumor burden. Moreover, histopathological analysis of the healthy tissues recovered after therapy, did not show any abnormalities. These new data are included in the revised manuscript on pages 11 & 14, and Fig EV2.

Second, we performed a toxicity study in wild type mice administered with various doses of the RGD4C/AAVP-*HSVtk*, then systemic tissues were harvested and serum levels of the lactate dehydrogenase (LDH) were evaluated at day seven post vector administration. BALB/c mice were injected with increasing vector doses: 2.5×10^9 TU, 1×10^{10} TU and 5×10^{10} TU. These doses were based on previous studies in mice, pet dogs, and on the doses used in the clinical study published by Krag et al. 2006 as follow:

Tumour-bearing Mice RGD4C/AAVP: 2×10^{12} TU/kg, equivalent of 5×10^{10} TU/mouse

Pet Dogs RGD4C/AAVP: 5×10^{12} TU/Dog = 5×10^{11} TU/kg, equivalent of 1×10^{10} TU/mouse

Cancer patients M13 max dose: 1×10^{11} TU/kg, is equivalent of 2.5×10^9 TU/mouse

We did not find any histopathologic changes in the healthy tissues, the LDH levels in the sera were normal, and no animal weight loss was noticed. These new data are now included in the revised manuscript on page 11 and Fig EV3.

Finally, we carried out *in vitro* treatments of various human normal cells, primary astrocytes, primary chondrocytes and primary fibroblasts from lung and skin. These cells showed low levels of expression of α_v and β_5 integrins, but no or barely detectable levels of β_3 . Importantly, this low integrin expression profile of normal cells, did not translate into detectable gene delivery by the RGD4C/AAVP vectors, alone or in the presence of TMZ. These data are now included in the revised version of this manuscript on pp. 11-12 and in Fig 6.

9. Were any studies carried out with primary GBM?

Response:

To comply with the reviewer, we have indeed carried out *in vivo* evaluation on primary GBM tumor cells implanted intracranially in immunodeficient mice.

First, we performed extensive work *in vitro* to screen primary GBM for their response to the RGD4C/AAVP before any *in vivo* investigation, as we abide by applying the principles of the 3Rs (Replacement, Reduction and Refinement). We used human primary GBM cells from different sources,

-023 cells from Portsmouth University and HSJD-GBM-001 from the Hospital Sant Joan de Deu Barcelona, Spain. Our findings show that all these human primary GBM cells express the integrin subunits α_v , β_3 and β_5 that are required for the formation of the heterodimer receptors $\alpha_v\beta_3$ and $\alpha_v\beta_5$. Then we showed that the RGD4C/AAVP vector efficiently delivered the *Luc* reporter gene to the all these primary cells, in a targeted manner. These data are reported in the revised manuscript on page 11 and in Fig 6.

We also included in this *in vitro* investigation two primary GBM stem cells (GSC), G26 and G166, from the collection of Dr Steven Pollard, expert in Neural Stem cells and Brain Cancer, from the Edinburgh Brain Cancer. We found that these GSC have high levels of α_v , β_3 and β_5 integrin expression, which translates into efficient and targeted gene delivery by the RGD4C/AAVP.

Moreover, gene delivery was enhanced by TMZ and subsequently the combination of RGD4C/AAVP-Grp78-HSVtk/GCV with TMZ induced the highest killing of these primary GBM stem cells. These data are included in the revised manuscript on pp. 12-13 and in Fig 7.

Next, *in vivo* experiments were carried out using intracranial primary GBM HSJD-GBM-001 established in nude mice, and showed that combination of RGD4C/AAVP-Grp78-HSVtk/GCV with TMZ resulted in the greatest i) tumor growth inhibition, ii) survival benefit of GBM-bearing mice, iii) reduction of Ki67 cell proliferation marker and iv) induction of apoptosis in primary GBM, iii), as compared to monotherapies of vector or TMZ alone. These findings are now included in the revised manuscript on pp. 13-14 and in Fig 8.

10. What was the mortality of the animals vs controls?

Response:

We would like to thank the reviewer for this point that was also raised by the other reviewers. Briefly, to comply with the reviewer, we performed additional experiments to evaluate the efficacy of treatments on the survival of tumor-bearing mice with either intracranial U87 GBM or intracranial primary GBM (HSJD-GBM-001). All the treatments tested: RGD4C/AAVP-Grp78-HSVtk vector/GCV alone, TMZ alone and combination of RGD4C/AAVP-Grp78-HSVtk/GCV plus TMZ improved the survival of both tumor-bearing mice with either U87 or primary GBM.

- Treatment with RGD4C/AAVP-Grp78-HSVtk/GCV exhibited better survival for both U87- and GBM-bearing mice when compared to treatment with the non-targeted AAVP/Grp78/GCV vector control.
- Importantly combination treatment with RGD4C/AAVP-Grp78-HSVtk/GCV and TMZ resulted in the highest survival benefit as compared to TMZ or RGD4C/AAVP-Grp78-HSVtk/GCV alone.
- Nicely, the overall trend of animal mortality was consistent between U87 and primary GBM and mortality was not biased against controls. So, we are specifically addressing the reviewer's comment.

These results are now included in the revised manuscript on pages 10 & 13 and Figs 4 & 8.

Referee #4

We would like to thank the reviewer for the critical review of our manuscript, we appreciate the comment "*The manuscript is well written and the claims are sustained by convincing data*".

We address her/his specific comments as follows:

- *The main limitation of manuscript is that no data regarding the safety and specificity of the treatment is provided.*

Response:

As also raised by another reviewer, we and independent groups have previously published a large body of data showing specificity and safety of the RGD4C/AAVP carrying various genes in rodents and pet dogs with natural tumors. A safety study using tumor-targeted M13 phages, parent of RGD4C/AAVP, was also completed in cancer patients. Moreover, to date in all the pre-clinical models tested, biodistribution of gene delivery by the RGD4C/AAVP vector generated gene expression in tumors exclusively without any detectable gene expression in the healthy tissues. In tumor-bearing mice, in addition to non-targeted AAVP, additional controls such as scrambled AAVP or mutated RGE/AAVP did not generate any gene delivery to tumors proving the tumor selectivity of the RGD4C ligand.

Briefly, regarding specificity studies in tumor-bearing mice, as we previously reported (Hajitou et al. 2006), biodistribution using bioluminescent imaging of luciferase in living tumor-bearing mice with vectors carrying the *Luc* reporter gene, showed specific expression of the *Luc* reporter in tumors with no expression detected in the healthy tissues, these data were confirmed with GFP vectors. In the same report (Hajitou et al. 2006), we reported an elegant study using micro-PET imaging of HSVtk with [¹⁸F]FEAU (2'-[¹⁸F]-fluoro-2'-deoxy-1-β-D-arabino-furanosyl-5-ethyl-uracil), radiolabelled substrate of the HSVtk enzyme and showed specific accumulation of [¹⁸F]FEAU in the tumors of mice and subsequently selective expression of the HSVtk, with no

accumulation of the [¹⁸F]FEAU in the healthy tissues. Histopathological analyses of the normal tissues removed from tumor-bearing mice treated by the same experimental protocol revealed no histopathologic abnormalities. Tandle et al, 2009, used the RGD4C/AAVP for targeted gene therapy with the cytokine, *TNF α* gene, and they reported specific expression of *TNF α* in the tumours following intravenous administration to tumor-bearing mice, with no *TNF α* detected in the healthy tissues.

A safety study using RGD4C/AAVP-*TNF α* was carried out in larger animals, under the direction of the National Cancer Institute of the USA elegantly demonstrated the safety and potential of this technology. Targeted RGD4C/AAVP was used to deliver *TNF α* to naturally occurring cancers diagnosed in dogs (Paoloni et al. PlosOne 2009). In this study, intravenous infusion of the targeted RGD4C/AAVP-*TNF α* generated selective accumulation of the vector particles in the tumours, with no vector detected in the healthy tissues, and this tumour accumulation of the vector translated in selective expression of *TNF α* in the tumours, exclusively. Dose escalation used 4×10^{11} , 8×10^{11} , 1×10^{12} , 5×10^{12} and 1×10^{13} vector TU/animal. The only significant adverse event observed occurred in one dog during the intravenous infusion of RGD4C/AAVP-*TNF α* in the highest dose cohort (1×10^{13} TU), where a single dose-limiting toxicity (DLT) was noted (Grade 3 hypersensitivity reaction); this event (nausea, tachycardia, and hypotension) was transient and resolved with minimal supportive care. Three additional dogs were entered into this dose cohort with no further toxicity observed. No maximally tolerated dose (MTD) was reached since the highest dose failed to result in any dose limiting toxicities. There were no clinically significant neurological, cardiac, respiratory, gastrointestinal, renal, or hematologic toxicities related to the treatment of the dogs entered in this phase of the study. Moreover, there were no delays in wound healing (surgical incision) detected, or febrile episodes associated with single-doses of RGD4C/AAVP-*TNF α* . These data suggest that RGD4C/AAVP-*TNF α* was safe following intravenous administration of a single dose of RGD4C/AAVP-*TNF α* to 10^{13} TU this dose is equivalent to the 5×10^{10} TU dose that we tested in our study in mice with GBM. Since no MTD was achieved, it is possible that higher doses of RGD4C/AAVP-*TNF α* may also be safely administered. It is noteworthy to mention that the cytokine *TNF α* has strong anti-tumor activity against various cancer types, but its clinical applicability has been hindered by its severe systemic toxicity. Targeted delivery by the RGD4C/AAVP has demonstrated potential to resolve the toxicity issue of *TNF α* .

In the clinical trial published by (Krag et al. Cancer Res. 2006), phage display libraries were injected to eight patients with stage IV cancer, including breast, melanoma and pancreas. The phage library, based on M13 phage (parent of the RGD4C/PAAV), was administered intravenously in order to identify tumor-targeted M13 phages. No serious clinical side effects, including allergic reactions, were observed with serial up to three intravenous infusions, despite the presence of anti-phage IgGs. In that trial, patients received 1.6×10^8 to 1.0×10^{11} TU/kg, equivalent of an approximate range of $\sim 1.0 \times 10^{10}$ to 1×10^{13} TU per patient. These phage doses were able to successfully target the tumors following intravenous administration. Immediate side effects during infusion were observed in only one patient during the second infusion. The side effect was grade 1 to 2 pain in the upper mid-back after 75% of the phage had been infused. The infusion was immediately stopped, and the pain fully subsided after 1 to 3 minutes. No other side effects attributable to the infusion were observed.

Moreover, to comply with the reviewer, we performed additional experiments as a toxicity study in immunocompetent wild-type mice administered with various doses of the RGD4C/AAVP-*HSVtk*, then systemic tissues were harvested and serum levels of the lactate dehydrogenase (LDH) were evaluated. BALB/c mice were injected with increasing doses: 2.5×10^9 TU, 1×10^{10} TU and 5×10^{10} TU. These doses were based on previous studies in mice, pet dogs, and on the doses used in the clinical study published by Krag et al. 2006 as follow:

Tumor-bearing Mice RGD4C/AAVP: = 2×10^{12} TU/kg, equivalent of 5×10^{10} TU/mouse
Pet Dogs RGD4C/AAVP: 5×10^{12} TU/Dog = 5×10^{11} TU/kg, equivalent of 1×10^{10} TU/mouse
Cancer patients M13 max dose: 1×10^{11} TU/kg, is equivalent of 2.5×10^9 TU/mouse

We did not observe any damage or abnormalities in the healthy tissues, the LDH measurements showed normal levels, and no animal weight loss was noticed. These new data are included in the revised manuscript on page 11 and Fig EV3.

• Although the targeting appears to be very specific for the 3 different human GBM cell lines analyzed, while mouse cells are not transduced, the study does not address the impact of the treatment on other human cell types. Indeed, the RGD4C ligand was selected to bind specifically the heterodimer $\alpha_v\beta_3$ integrin in human cells which is expressed in a wide variety of cells and not only GBM. What is the effect of the treatment on other human cell types? This is an outstanding issue since this treatment could result in an extreme toxicity (potentially lethal) to the patients. The Authors should address this outstanding issue experimentally on a large panel of human primary cells and cell lines.

Response:

The RGD4C ligand binds mainly to $\alpha_v\beta_3$ but also to a lesser level to the $\alpha_v\beta_3$ heterodimer. Various integrin heterodimers can be found in wide variety of human cells; however, both $\alpha_v\beta_3$ and $\alpha_v\beta_5$ are highly restricted and typically overexpressed on cancer cells and tumor vasculature (Arap et al, 1998; Hemminki et al, 2001). Additionally, tumor metastasis, rheumatoid arthritis and osteoporosis are also on the list of pathologies in which $\alpha_v\beta_3$ plays a key role (Kumar 2003).

In Human biopsies, the $\alpha_v\beta_3$ integrin is widely expressed on blood vessels of human tumor biopsy samples but not on vessels of biopsies from normal tissues; the distribution of $\alpha_v\beta_3$ in human is highly restricted, with expression on activated endothelium, activated vascular smooth muscle and tumors. Besides this, the $\alpha_v\beta_3$ has been shown to have relatively limited cellular distribution in humans quiescent tissues; apart from its expression at high levels in the inflamed synovial tissues of rheumatoid arthritis patients (Koch 2000, Pap et al, 2000), $\alpha_v\beta_3$ is absent or minimally, or barely detectable on endothelial cells (Van De Wiele et al, 2002), some B-cells, platelets, monocytes, intestinal cells, and smooth muscle cells, as well as a small percentage of activated leukocytes, macrophages, and osteoclasts.

Integrin $\alpha_v\beta_5$ is often found in the same pathological contexts as $\alpha_v\beta_3$, but can also be found in fibroblasts (Gamble et al, 2010).

Therefore, to comply with the reviewer we sought to carry out an *in vitro* investigation comprising diverse human primary cells with various integrin expression patterns, from no integrin expression to the low expression reported above, in order to seek whether the minimal expression on normal primary cells is enough to generate successful gene delivery by the RGD4C/AAVP. First, because $\alpha_v\beta_5$ is also found expressed on normal fibroblasts, we examined two types of human normal primary fibroblasts from skin and lung. We found that these cells do not or barely express the β_3 integrin but have low levels of expression of α_v and β_5 subunits, compared to primary GBM. Importantly however, this low integrin expression profile did not translate in any detectable gene delivery by the RGD4C/AAVP vector, alone or even in the presence of TMZ. Moreover, to further comply with the reviewer, we included additional normal human primary cells, astrocytes and chondrocytes. While the astrocytes showed some integrin expression of α_v and β_5 , with low expression of β_3 , this integrin profile did not permit gene delivery when cells were treated with the RGD4C/AAVP alone or in combination with TMZ. Immunostaining of the chondrocytes did not show any expression of these integrins, and no subsequent gene delivery was detected after addition of the RGD4C/AAVP.

To recap, our data show no expression or minimal expression of these integrins in some human normal cells, which does not permit any gene expression by the RGD4C/AAVP vector. The integrin expression in GBM is substantially higher than that in human normal cells. Targeting depends on differences in the levels of expression of the target receptors. Moreover, it is important to mention that RGD4C/AAVP-*Grp78* should allow dual GBM targeting, by RGD4C ligand (entry) and *Grp78* promoter (transcriptional targeting) which should result in increased tumor selectivity and safety of this gene therapy treatment. Thus, even if this minimal integrin expression on normal cells allows phage internalisation through binding of the RGD4C to integrins, the promoter *Grp78* will ensure selectivity, in particular that TMZ did not induce any gene expression by RGD4C/AAVP in these normal cells. It is correct that non-cancerous human immortalized cell lines, e.g. Human Embryonic Kidney HEK293 cells, could have higher expression of these integrins; however, unlike normal primary cells, these cell lines exhibit a certain degree of transformation because of their immortalized phenotype.

Finally, any clinical trial will commence with a phase-1 trial for a safety study of the treatment. Phage has a historic safety profile in human, over many years to treat infectious diseases (Asavarut and Hajitou 2014). We would stress that each element such as intravenous M13 phage administration, RGD peptides and Grp78 promoter have been established as safe.

All these data on normal primary cells are now included in the revised version of this manuscript on pp. 11-12 and in Fig 6.

References

- Arap W, Pasqualini R, Ruoslahti E (1998) Cancer treatment by targeted drug delivery to tumor vasculature in a mouse model. *Science* 279: 377–380
- Asavarut P, Hajitou A (2014) The phage revolution against antibiotic resistance. *The Lancet Infectious Diseases* 14
- Black ME, Kokoris MS, Sabo P (2001) Herpes simplex virus-1 thymidine kinase mutants created by semi-random sequence mutagenesis improve prodrug-mediated tumor cell killing. *Cancer Res* 61: 3022-3026
- Culver KW, Ram Z, Wallbridge S, Ishii H, Oldfield EH, Blaese RM (1992) *In vivo* gene therapy with retroviral vector-producer cells for treatment of experimental brain tumors. *Science*: 256: 1550–2.
- Dubos RJ, Straus JH, Pierce C (1943) The Multiplication of Bacteriophage in Vivo and Its Protective Effect against an Experimental Infection with Shigella Dysenteriae. *J Exp Med* 78: 161-168
- Gamble LJ, Borovjagin AV, Matthews QL (2010) Role of RGD-containing ligands in targeting cellular integrins: Applications for ovarian cancer virotherapy. *Exp Ther Med* 1: 233–240.
- Geier MR, Trigg ME, Merrill CR (1973) Fate of bacteriophage lambda in non-immune germ-free mice. *Nature* 246: 221–223.
- Gladson CL, Cheresh DA (1991) Glioblastoma expression of vitronectin and the alpha v beta 3 integrin. Adhesion mechanism for transformed glial cells. *J Clin Invest* 88: 1924-1932.
- Grignet-Debrus C, Cool V, Baudson N, Velu T, Calberg-Bacq CM (2000) The role of cellular- and prodrug-associated factors in the bystander effect induced by the Varicella zoster and Herpes simplex viral thymidine kinases in suicide gene therapy. *Cancer Gene Ther* 7: 1456-1468.
- Hajitou A, Trepel M, Lilley CE, Soghomonyan S, Alauddin MM, Marini FC, 3rd, Restel BH, Ozawa MG, Moya CA, Rangel R et al (2006) A hybrid vector for ligand-directed tumor targeting and molecular imaging. *Cell* 125: 385-398.
- Hamel W, Magnelli L, Chiarugi VP, Israel MA (1996) Herpes simplex virus thymidine kinase/ganciclovir-mediated apoptotic death of bystander cells. *Cancer Res* 56: 2697–2702.
- Hemminki A, Belousova N, Zinn KR, Liu B, Wang M, Chaudhuri TR, Rogers BE, Buchsbaum DJ, Siegal GP, Barnes MN et al (2001) An adenovirus with enhanced infectivity mediates molecular chemotherapy of ovarian cancer cells and allows imaging of gene expression. *Mol Ther* 4: 223–231
- Ho IA, Lam PY, Hui KM (2004) Identification and characterization of novel human glioma-specific peptides to potentiate tumor-specific gene delivery. *Hum Gene Ther* 15: 719-732.
- Ho IA, Hui KM, Lam PY (2010) Isolation of peptide ligands that interact specifically with human glioma cells. *Peptides* 31: 644-650.
- Huang T-Y, Tsai T-H, Hsu C-W, Hsu Y-C (2010) Curcuminoids Suppress the Growth and Induce Apoptosis through Caspase-3-Dependent Pathways in Glioblastoma Multiforme (GBM) 8401 Cells. *J Agric Food Chem* 58: 10639–10645.

- Kia A, Przystal JM, Nianiaris N, Mazarakis ND, Mintz PJ, Hajitou A (2012) Dual systemic tumor targeting with ligand-directed phage and Grp78 promoter induces tumor regression. *Mol Cancer Ther* 11: 2566-2577.
- Kim B, Kim HS, Jung EJ, Lee JY, B KT, Lim JM, Song YS (2016) Curcumin induces ER stress-mediated apoptosis through selective generation of reactive oxygen species in cervical cancer cells. *Mol Carcinog* 55: 918-928.
- Klinger NV, Mittal S (2016) Therapeutic Potential of Curcumin for the Treatment of Brain Tumors. *Oxid Med Cell Longev* 2016: 9324085.
- Koch AE (2000) The role of angiogenesis in rheumatoid arthritis: recent developments. *Ann Rheum Dis* 59 (Suppl 1):i 65-i71
- Krag DN, Shukla GS, Shen GP, Pero S, Ashikaga T, Fuller S, Weaver DL, Burdette-Radoux S, Thomas C (2006) Selection of tumor-binding ligands in cancer patients with phage display libraries. *Cancer Res* 66: 7724-7733.
- Kumar CC (2003) Integrin alpha v beta 3 as a therapeutic target for blocking tumor-induced angiogenesis. *Curr Drug Targets* 4:123-131
- Li J, Zhang Q, Pang Z, Wang Y, Liu Q, Guo L, Jiang X (2012) Identification of peptide sequences that target to the brain using in vivo phage display. *Amino Acids* 42: 2373-81.
- Mandic A, Hansson J, Linder S, Shoshan MC (2003) Cisplatin induces endoplasmic reticulum stress and nucleus-independent apoptotic signaling. *J Biol Chem* 278: 9100-9106.
- Paoloni MC, Tandle A, Mazcko C, Hanna E, Kachala S, Leblanc A, Newman S, Vail D, Henry C, Thamm D et al (2009) Launching a novel preclinical infrastructure: comparative oncology trials consortium directed therapeutic targeting of TNFalpha to cancer vasculature. *PLoS One* 4: e4972
- Park W, Amin ARM, Chen ZG, Shin DM (2013) New perspectives of curcumin in cancer prevention. *Cancer Prev Res (Phila)* 6: 387-400.
- Pap T, Gay R, Gay S (2000) Mechanisms of joint destruction. In: Firestein GS, Panayi GS, Wollheim FA, editors. *Rheumatoid Arthritis. New Frontiers in Pathogenesis and Treatment*. Oxford University Press; Oxford: 2000. pp. 189-199
- Pollard SM (2013) In vitro expansion of fetal neural progenitors as adherent cell lines. *Methods Mol Biol* 1059: 13-24.
- Schnell O, Krebs B, Wagner E, Romagna A, Beer AJ, Grau SJ, Thon N, Goetz C, Kretzschmar HA, Tonn JC et al (2008) Expression of integrin alphavbeta3 in gliomas correlates with tumor grade and is not restricted to tumor vasculature. *Brain Pathol* 18: 378-386.
- Sun C, Han C, Jiang Y, Han N, Zhang M, Li G, Qiao Q (2017) Inhibition of GRP78 abrogates radioresistance in oropharyngeal carcinoma cells after EGFR inhibition by cetuximab. *PLoS One* 12: e0188932.
- Tandle A, Hanna E, Lorang D, Hajitou A, Moya CA, Pasqualini R, Arap W, Adem A, Starker E, Hewitt S et al (2009) Tumor vasculature-targeted delivery of tumor necrosis factor-alpha. *Cancer* 115: 128-139
- Trepel M, Arap W, Pasqualini R (2001) Modulation of the immune response by systemic targeting of antigens to lymph nodes. *Cancer Res* 61: 8110-8112.
- Trepel M, Stoneham CA, Eleftherohorinou H, Mazarakis ND, Pasqualini R, Arap W, Hajitou A (2009) A heterotypic bystander effect for tumor cell killing after adeno-associated virus/phage-mediated, vascular-targeted suicide gene transfer. *Mol Cancer Ther* 8: 2383-2391.

Van De Wiele C, Oltenfreiter R, De Winter O, Signore A, Slegers G, Dierckx RA (2002) Tumour angiogenesis pathways: related clinical issues and implications for nuclear medicine imaging. *Eur J Nucl Med Mol Imaging* 29: 699–709

Yata T, Lee EL, Suwan K, Syed N, Asavarut P, Hajitou A (2015) Modulation of extracellular matrix in cancer is associated with enhanced tumor cell targeting by bacteriophage vectors. *Mol Cancer* 14:110.

2nd Editorial Decision

13 August 2018

Thank you for the submission of your revised manuscript to EMBO Molecular Medicine. We have now received the enclosed reports from the referees that were asked to re-assess it. As you will see, while referee 4 is now satisfied, referees 1 and 3 are not yet and both insist on revisions that they asked before but were not performed.

I am afraid that we agree with referees 1 and 3 and would encourage you to perform the requested revisions for publication as EMBO Molecular Medicine focuses on translational relevance. I am happy to extend the usual 2-weeks revision time to allow for extra work to be done.

Should you decide not to perform these and rather prefer to seek publication elsewhere, please do let us know as soon as possible.

I look forward to reading a new revised version of your manuscript as soon as possible.

***** Reviewer's comments *****

Referee #1 (Remarks for Author):

The authors have satisfied the majority of concerns raised during the revision.

Although, one major point has not been fully addressed: they didn't perform *in vivo* experiments using human GBM CSCs. They obtained GBM CSCs from Pollard laboratory whose expertise in CSCs characterization is well known in the field.

It is unlikely that these cells "were not consistent in inducing tumors in immune-deficient mice".

Although GBM is a tumor with high heterogeneity among patients, generally GBM CSCs are highly tumorigenic in a short period after orthotopic cell implantation.

The authors should better address this point also to take in account GBM CSC resistance to TMZ *in vivo*.

Minor points:

Fig1C: They should provide not cropped images of Western Blots.

Referee #3 (Remarks for Author):

The following comments to the authors relate only to the investigator's response to this reviewer (#3), with the following point-by-point same numbering as in the original review:

1. Adequate response.
2. The issue of the use of TMZ is important. The investigators provide alternatives, but for reasons that are unclear, refuse to put the alternative data into the manuscript and only have inserted a brief discussion about this issue in the revised manuscript (p15). The Curcumin data, at least at 40 uM, adds significantly to dealing with this issue, and the methods, results and figure should be included in the revised manuscript
3. The issue of how much phage crosses the blood-brain barrier and reaches the tumor is critical. The investigators provide some data to rebut the review concern, but have not added it to the revised

manuscript. The revised manuscript should include the methods, results and graphs with the data, including statistical analysis etc.

4. Immunogenicity is a critical issue for all gene therapy approaches, and had been a major concern (and cause of death) in human in vivo gene therapy studies. Despite the many arguments provided by the investigators, they choose to use immunodeficient rodents in their studies. While this allows for focus on the therapy without confounding effects of the immune system, it ignores the very real possibility of the immune system limiting efficacy and the immune system inducing toxicity. To make this study valuable, studies should be carried out in immunocompetent animals.

5. The statement of data helps, but should be enhanced regarding the implications of the data relating to how much and where is the integrins expressed? If normal endothelial and glial cells express the integrin, is this a problem for off-target toxicity? It is not necessary to do additional experiments, but the statements regarding what % glioblastoma and what normal cells express the integrins should be enlarged to discuss the implications for both efficacy and toxicity.

6. Adequate response.

7. The investigators agree that the data is important, but do not add the data to the paper. Off target, other organ data is critical in regard to the feasibility of moving this therapy to humans.

8. The added data is useful regarding potential toxicity.

9. The added data is useful regarding primary tumors.

10. The added data is useful regarding mortality.

Referee #4 (Remarks for Author):

the Authors answered satisfactorily to the concerns raised by this reviewer

2nd Revision - authors' response

20 December 2018

We are pleased to submit a second revised manuscript to address the additional reviewers' concerns. We agree with the importance of the experiments requested by the reviewers and we are happy with the new findings, which should definitely strengthen this manuscript. We have addressed the reviewers' comments point-by-point in the text below. We have also highlighted the changes, in yellow, in the revised manuscript to make them clearly visible to the reviewers.

Referee #1

1. The authors have satisfied the majority of concerns raised during the revision.

Although, one major point has not been fully addressed: they didn't perform in vivo experiments using human GBM CSCs. They obtained GBM CSCs from Pollard laboratory whose expertise in CSCs characterization is well known in the field.

It is unlikely that these cells "were not consistent in inducing tumors in immune-deficient mice".

Although GBM is a tumor with high heterogeneity among patients, generally GBM CSCs are highly tumorigenic in a short period after orthotopic cell implantation.

The authors should better address this point also to take in account GBM CSC resistance to TMZ in vivo.

Response:

We thank the reviewer for the comment "*The authors have satisfied the majority of concerns raised during the revision*", and agree that in vivo efficacy on GBM CSCs is important to address.

Therefore, we have carried out various experiments.

First and as mentioned by the reviewer, GBM is a highly heterogeneous tumor that contains GBM stem cell sub-population. In our first revision we used the HSJD-GBM-001 from the collection of

Dr Carcaboso to assess in vivo efficacy on primary GBM that we cultured in vitro in serum-free medium to grow spheroids before implantation into the brain of mice. Consequently, we tested the HSJD-GBM-001 spheres for the presence of CSCs by using antibodies against Sox-2, Nestin and CD133 stem cell markers and compared side-by-side with the G26 primary GBM cells received from Dr Pollard. The data showed that HSJD-GBM-001 spheroids express high levels of these stem cell markers (65.7% Sox-2⁺ cells, 62.7% of Nestin⁺ cells and 93.6% CD133⁺ cells) proving that these GBM spheres contain high percentage of stem cells. Whereas the G26 cells had 13.4% Sox-2⁺ cells, 45.7% Nestin⁺ cells and 67% CD133⁺ cells. Moreover, immunohistochemistry of intracranial HSJD-GBM-001 tumors for stem cell markers showed high levels of CSCs. Next, analyses of CSC in intracranial HSJD-GBM-001 recovered after therapy showed clear reduction of CD133⁺ CSC cells following treatment with the vector RGD4C/AAVP-Grp78-HSVtk + GCV, compared to TMZ treatment. Then, remarkably, combination treatment with vector and TMZ resulted in suppression of the CD133⁺ cells. These data are now included in the new version of the manuscript in Fig 6A and Fig 8D, and on pp.11-14 of the Results Section. We have also amended the discussion on page 18. Moreover, we have included additional data showing RGD4C/AAVP-Grp78HSVtk vector homing to HSJD-GBM-001 tumors but not to healthy tissues, in Fig EV3D and page 14.

We would like to mention that these findings are consistent with the therapy data on Fig 8 showing that TMZ alone was not able to eradicate the tumors and that combining with HSVtk/GCV achieved better effect, which also could be explained by the efficacy of CSC transduction reported in Fig 6.

2. Minor points:

Fig1C: They should provide not cropped images of Western Blots.

Response: we have complied with the reviewer request and replaced the cropped images of Western Blots in Fig 1C.

Referee #3

2. The issue of the use of TMZ is important. The investigators provide alternatives, but for reasons that are unclear, refuse to put the alternative data into the manuscript and only have inserted a brief discussion about this issue in the revised manuscript (p15). The Curcumin data, at least at 40 uM, adds significantly to dealing with this issue, and the methods, results and figure should be included in the revised manuscript

Response:

We have complied with the reviewer and included this data in the manuscript, in the Results Section on page 7, in the Method section on page 20 and in Fig EV2. We also added a statement in the Discussion on pages 16 & 17.

3. The issue of how much phage crosses the blood-brain barrier and reaches the tumor is critical. The investigators provide some data to rebut the review concern, but have not added it to the revised manuscript. The revised manuscript should include the methods, results and graphs with the data, including statistical analysis etc.

Response:

We have complied with the reviewer and added the data in the Results Section of the manuscript on page 9 and in Fig 3D. We have also revised the methods for the vector homing to tumors on page 22 to provide more information, and amended the discussion on page 18.

4. Immunogenicity is a critical issue for all gene therapy approaches, and had been a major concern (and cause of death) in human in vivo gene therapy studies. Despite the many arguments provided by the investigators, they choose to use immunodeficient rodents in their studies. While this allows for focus on the therapy without confounding effects of the immune system, it ignores the very real possibility of the immune system limiting efficacy and the immune system inducing toxicity. To make this study valuable, studies should be carried out in immunocompetent animals.

Response:

We would like to apologise that we did not clearly get this point in the initial revision. We thank the reviewer for clarifying this important point that has further strengthened this study. We have now complied with the reviewer and performed experiments in immunocompetent C57Black/6J mice with established intracranial GL261 GBM model. We chose this isogenic tumor model because it is the most recognized murine GBM that has been extensively used for preclinical testing of therapeutic approaches for GBM. We first confirmed that the GL261 cells express the integrin receptors, required for transduction by the RGD4C/AAVP vector. Then our in vivo studies clearly established the anti-tumor effects of the RGD4C/AAVP-Grp78-HSVtk plus GCV against the intracranial GL261 GBM; interestingly, combination of vector with TMZ resulted in complete suppression of tumor growth. In this study, vector administrations were repeated three times, safely and efficiently, consistent with our previous findings in immunocompetent animals and in vaccinated mice. Phage-based particles are known to be immunogenic, but this feature can be modulated through targeting itself (Trepel et al, 2001).

All these data are now included in the revised manuscript in Fig 9, as well as in the Results Section on pages 14 & 15, in the Discussion on page 19, and in the Method Section.

5. The statement of data helps, but should be enhanced regarding the implications of the data relating to how much and where is the integrins expressed? If normal endothelial and glial cells express the integrin, is this a problem for off-target toxicity? It is not necessary to do additional experiments, but the statements regarding what % glioblastoma and what normal cells express the integrins should be enlarged to discuss the implications for both efficacy and toxicity.

Response:

We apologise to the reviewer for this confusion. In the previous rebuttal letter, we meant expression of this integrin in the activated endothelial cells that form the angiogenic blood vessels in tumors; we also used glial cells to mean the glial-derived tumor cells. In other words we meant angiogenic endothelial and glial tumor cells present in the tumor tissue. Consequently, we have modified the statement that we included in the discussion, on page 16 as follows:

“Previous studies reported that $\alpha_v\beta_3$ integrin can be expressed in up to 55% of human glioblastoma, depending on the tumor grade (Schittenhelm et al., 2013; Schnell et al. 2008). The $\alpha_v\beta_3$ expression was significantly higher in human GBMs than in low-grade gliomas and was seen on both activated endothelial cells and glial tumor cells existing within glioblastomas (Schittenhelm et al., 2013; Schnell et al. 2008). In contrast, expression of $\alpha_v\beta_3$ is barely detectable on normal human endothelial cells (Gamble et al, 2010; Kumar 2003; Koch 2000; Pap et al, 2000; Van De Wiele et al, 2002), and is also not expressed on glia or neurons in normal adult cortex and cerebral white matter (Gladson and Cheresh 2001).

Previously, we and independent groups reported that the low level of expression of $\alpha_v\beta_3$ integrin in normal endothelial cells did not lead to phage localisation in the blood vessels of normal tissues in mice as well as in large animals e.g. pet dogs (Hajitou et al. Cell 2006; Tandle et al. Cancer 2009; Paoloni et al. PlosOne 2009). We also reported absence of gene delivery (GFP, Luc and HSVtk expression) by the RGD4C/AAVP in normal blood vessels and healthy tissues including the brain (Hajitou et al. 2006, Tandle et al. 2009, Paoloni et al. 2009).

To further comply with the reviewer and as also requested by another reviewer, we tested the vector in normal human glial cells (primary astrocytes). The RGD4C ligand binds mainly to $\alpha_v\beta_3$, but also to a lesser level to the $\alpha_v\beta_5$ heterodimer. We found that normal human astrocytes have very low levels of α_v and β_5 integrins with barely detectable expression of β_3 , compared to primary GBM. Importantly however, this low integrin expression profile did not lead to gene delivery by the RGD4C/AAVP-Grp78 vector. Moreover, addition of TMZ did not induce any gene expression from RGD4C/AAVP-Grp78 in these normal glial cells. It is noteworthy to mention that RGD4C/AAVP-Grp78 vector has the advantage to allow dual GBM targeting, by RGD4C ligand (entry) and Grp78 promoter (transcriptional targeting) which should result in increased tumor selectivity and safety of this gene therapy treatment. Thus, even if this minimal integrin expression on normal cells allows vector entry into cells through binding of the RGD4C to integrins, the tumor specific Grp78 promoter which is not active in normal cells/tissues won't allow gene delivery into these normal cells.

These data on glial cells are shown in the revised manuscript in Fig 7 and in page 12 of the Results Section. We also added a statement in the Discussion Section on page 16.

7. The investigators agree that the data is important, but do not add the data to the paper. Off target, other organ data is critical in regard to the feasibility of moving this therapy to humans.

Response:

In the original manuscript, we reported that no vector was detected in the healthy brain in Fig 3B that shows accumulation of the RGD/AAVP-Grp78-HSVtk vector in the tumor tissue but not in the adjacent normal brain. Now, we have further complied with the reviewer and carried out immunostaining experiments on sections from vital healthy tissues (liver, kidney, heart and lung) recovered from mice following intravenous administration of vector. We used an antibody against the capsid of RGD4C/AAVP-Grp78-HSVtk vector. Microscopic analyses of tissue sections didn't show any presence of the RGD4C/AAVP-Grp78-HSVtk vector in all the normal tissues tested; whereas, tumors were positive for the presence of vector in mice receiving RGD4C/AAVP-Grp78-HSVtk alone or combination of RGD4C/AAVP-Grp78-HSVtk with TMZ. These findings are consistent with previous studies from our group and reports from independent groups. These data are now included in the revised version of the manuscript in the Results Section on page 14, in Fig EV3D. We also added a statement in the Discussion Section on page 18.

3rd Editorial Decision

22 January 2019

Thank you for the submission of your revised manuscript to EMBO Molecular Medicine. We have now received the enclosed reports from the referees that were asked to re-assess it. As you will see the reviewers are now supportive and I am pleased to inform you that we will be able to accept your manuscript pending minor editorial amendments.

***** Reviewer's comments *****

Referee #1 (Remarks for Author):

The Authors answered satisfactorily to the concerns raised during the second revision, providing the required experimental data about the in vivo efficacy of the proposed therapy in a orthotropic model of GBM xenotransplant using primary GBM tumor initiating cells. For this reason, I suggest that this manuscript is now suitable for publication.

Referee #3 (Remarks for Author):

None

3rd Revision - authors' response

1 February 2019

(The authors made the requested editorial changes.)

Corresponding Author Name: Amin Hajitou

Manuscript Number: EMM-2017-08492